# Bringing it all together: Science priorities for improved understanding of Earth system change and to support international climate policy.

Colin G. Jones[1], Fanny Adloff[2], Ben B. B. Booth[3], Peter M. Cox[4], Veronika Eyring[5,6], Pierre Friedlingstein[7,8], Katja Frieler[9], Helene T. Hewitt[3], Hazel A. Jeffery[1], Sylvie Joussaume[10], Torben Koenigk[11,12], Bryan N. Lawrence[13], Eleanor O'Rourke[14], Malcolm J. Roberts[3], Benjamin M. Sanderson[15], Roland Séférian[16], Samuel Somot[16], Pier Luigi Vidale[13], Detlef van Vuuren[17,18], Mario Acosta[19], Mats Bentsen[20,21], Raffaele Bernardello[19], Richard Betts[3,22], Ed Blockley[3], Julien Boé[23], Tom Bracegirdle[24], Pascale Braconnot[10], Victor Brovkin[25], Carlo Buontempo[26], Francisco Doblas-Reyes[19,27], Markus Donat[19], Italo Epicoco[28,29], Pete Falloon[3,30], Sandro Fiore[31], Thomas Frölicher[32,33], Neven S. Fučkar[34,35], Matthew J. Gidden[36], Helge F. Goessling[37], Rune Grand Graversen[38], Silvio Gualdi[39], José M. Gutiérrez[40], Tatiana Ilyina[41], Daniela Jacob[42], Chris D. Jones[3,43], Martin Juckes[1,44], Elizabeth Kendon[3,43], Erik Kjellström[11], Reto Knutti[45], Jason Lowe[3,46], Matthew Mizielinski[3], Paola Nassisi[28], Michael Obersteiner[47], Pierre Regnier[48], Romain Roehrig[16], David Salas y Mélia[16], Carl-Friedrich Schleussner[49], Michael Schulz[50], Enrico Scoccimarro[39], Laurent Terray[23], Hannes Thiemann[51], Richard A. Wood[3], Shuting Yang[52], Sönke Zaehle[53]

[1]National Centre for Atmospheric Science, University of Leeds, Leeds, UK
[2]ESMO International Project Office, German Climate Computing Center (DKRZ), Hamburg, Germany
[3]Met Office Hadley Centre, Exeter, UK
[4]Global Systems Institute, University of Exeter, Exeter UK
[5]Deutsches Zentrum für Luft- und Raumfahrt (DLR), Institut für Physik der Atmosphäre, Oberpfaffenhofen, Germany
[6]University of Bremen, Institute of Environmental Physics (IUP), Bremen, Germany
[7]Faculty of Environment, Science and Economy, University of Exeter, Exeter, UK
[8]Laboratoire de Météorologie Dynamique, Institut Pierre-Simon Laplace, CNRS, École Normale Supérieure, Université PSL, Sorbonne Université, École Polytechnique, Paris, France
[9]Potsdam Institute for Climate Impact Research, Potsdam, Germany
[10]Laboratoire des Sciences du Climat et de l'Environnement, Institut Pierre Simon Laplace, CNRS, CEA, Université Versailles Saint-Quentin, Université Paris-Saclay, Gif-sur-Yvette, France
[11]Swedish Meteorological and Hydrological Institute, Norrköping, Sweden
[12]Bolin Centre for Climate Research, Stockholm University, Stockholm, Sweden
[13]National Centre for Atmospheric Science, Department of Meteorology, University of Reading, Reading, UK
[14]CMIP International Project Office, ECSAT, Harwell Science & Innovation Campus, UK
[15]CICERO, Oslo, Norway
[16]CNRM, Université de Toulouse, Météo-France, CNRS, France
[17]PBL Netherlands Environmental Assessment Agency, the Hague, the Netherlands
[18]Copernicus Institute of Sustainable Development, Utrecht University, Utrecht, the Netherlands
[19]Barcelona Supercomputing Center, Barcelona, Spain
[20]NORCE Norwegian Research Centre, Bergen, Norway
[21]Bjerknes Centre for Climate Research, Bergen, Norway
[22]University of Exeter, Exeter, UK
[23]CECI, CNRS/Cerfacs, Toulouse, France
[24]British Antarctic Survey, Cambridge, UK
[25]Max Planck Institute for Meteorology, Bundesstr. 53, 20146, Hamburg, Germany
[26]ECMWF, Robert Schuman Platz, Bonn, Germany
[27]ICREA, Barcelona, Spain
[28]Euro-Mediterranean Center on Climate Change Foundation, Lecce, Italy
[29]University of Salento, Lecce, Italy
[30]School of Biology, Life Sciences Building, University of Bristol, UK
[31]University of Trento, Trento, Italy
[32]Climate and Environmental Physics, University of Bern, Switzerland
[33]Oeschger Centre for Climate Change Research, University of Bern, Switzerland
[34]Earth Sciences Department, Barcelona Supercomputing Center, Barcelona, Spain
[35]Environmental Change Institute, University of Oxford, Oxford, UK
[36]International Institute for Applied Systems Analysis, Laxenburg, Austria
[37]Alfred Wegener Institute, Helmholtz Centre for Polar and Marine Research, Bremerhaven, Germany
[38]Department of Physics and Technology, University of Tromsø, Norway
[39]Fondazione CMCC, Bologna, Italy
[40]Instituto de Física de Cantabria, CSIC - Universidad de Cantabria, Santander, Spain

[41]Universität Hamburg, Helmholtz-Zentrum Hereon, Max Planck Institute for Meteorology, Germany
[42]Climate Service Center Germany (GERICS), Helmholtz-Zentrum HEREON GmBh, Hamburg, Germany
[43]School of Geographical Sciences, University of Bristol, UK
[44]Science and Technology Facilities Council, Didcot, Oxfordshire, UK
[45]Institute for Atmospheric and Climate Science, ETH Zurich, Zurich, Switzerland
[46]Priestley Centre, University of Leeds, Leeds, UK
[47]Environmental Change Institute, University of Oxford, Oxford UK
[48]Department of Geoscience, Environment & Society-BGEOSYS, Université Libre de Bruxelles, Belgium
[49]Humboldt University, Berlin, Germany
[50]Norwegian Meteorological Institute, Oslo, Norway
[51]German Climate Computing Center (DKRZ), Hamburg, Germany
[52]National Centre for Climate Research, Danish Meteorological Institute, Copenhagen, Denmark
[53]Biogeochemical Signals Department, Max Planck Institute for Biogeochemistry, Jena, Germany
Correspondence to: Colin Jones (colin.jones@metoffice.gov.uk)
**Abstract.** We review how the international modelling community, encompassing Integrated Assessment models, global and
regional Earth system and climate models, and impact models, have worked together over the past few decades, to advance
understanding of Earth system change and its impacts on society and the environment, and thereby support international
climate policy. We go on to recommend a number of priority research areas for the coming decade, a timescale that
encompasses a number of newly starting international modelling activities, as well as the IPCC 7th Assessment Report
(AR7) and the 2nd UNFCCC Global Stocktake. Progress in these priority areas will significantly advance our understanding
of Earth system change and its impacts, increasing the quality and utility of science support to climate policy.

We emphasize the need for continued improvement in our understanding of, and ability to simulate, the coupled Earth
system and the impacts of Earth system change. There is an urgent need to investigate plausible pathways and emission
scenarios that realize the Paris Climate Targets. For example, pathways that overshoot 1.5°C or 2°C global warming, before
returning to these levels at some later date. Earth System models need to be capable of thoroughly assessing such warming
overshoots, in particular, the efficacy of mitigation measures, such as negative $CO_2$ emissions, in reducing atmospheric $CO_2$
and driving global cooling. An improved assessment of the long-term consequences of stabilizing climate at 1.5°C or 2°C
above pre-industrial temperatures is also required. We recommend Earth system models run overshoot scenarios in $CO_2$-
emission mode, to more fully represent coupled climate - carbon cycle feedbacks and, wherever possible, interactively
simulate other key Earth system phenomena at risk of rapid change during overshoot. Regional downscaling and impact
models should use forcing data from these simulations, so impact and regional climate projections cover a more complete
range of potential responses to a warming overshoot. An accurate simulation of the observed, historical record remains a
fundamental requirement of models, as does accurate simulation of key metrics, such as the Effective Climate Sensitivity
and the Transient climate response to cumulative carbon emissions. For adaptation, a key demand is improved guidance on
potential changes in climate extremes and the modes of variability these extremes develop within. Such improvements will
most likely be realized through a combination of increased model resolution, improvement of key model parameterizations,
enhanced representation of important Earth system processes, combined with targeted use of new Artificial Intelligence (AI)
and Machine Learning (ML) techniques. We propose a deeper collaboration across such efforts over the coming decade.

With respect to sampling future uncertainty, increased collaboration between approaches that emphasize large model
ensembles and those focussed on statistical emulation is required. We recommend an increased focus on High Impact Low
Likelihood (HILL) outcomes. In particular, the risk and consequences of exceeding critical tipping points during a warming
overshoot and the potential impacts arising from this. For a comprehensive assessment of the impacts of Earth system
change, including impacts arising directly as a result of climate mitigation actions, it is important spatially detailed,
disaggregated information used to generate future scenarios in Integrated Assessment Models are available for use in impact
models. Conversely, methods need to be developed that enable potential societal responses to projected Earth system change
to be incorporated into scenario development.

The new models, simulations, data, and scientific advances, proposed in this article will not be possible without long-term
development and maintenance of a robust, globally connected infrastructure ecosystem. This system must be easily
accessible and useable by modelling communities across the world, allowing the global research community to be fully
engaged in developing and delivering new scientific knowledge to support international climate policy.

## 1    Introduction

Given the rapidly developing climate crisis, and the negative consequences for planetary habitability and human well-being,
there is an increasing need for accurate, reliable, and actionable information encompassing the full spectrum of climate risk.
This information is required at global to local scales, near to long timescales, and needs to be tailored to inform critical
decision-making related to climate change mitigation and adaptation (e.g., in the context of UNFCCC negotiations, the UN
Global Stocktake, IPCC assessments, and the World Adaptation Science Program; WASP), as well as the growing needs of
climate service providers. Over the past few decades, coordinated by the World Climate Research Program (WCRP), the
international modelling community has worked together to contribute simulations, data and knowledge to support decision
making, in particular the cyclical IPCC Assessment Reports (AR). This has been achieved through a suite of interconnected
modelling projects and initiatives, with the most important of these listed in Table 1, along with project acronyms and
primary citations. Meehl (2023) discusses the synergistic interaction between climate science (particularly Global Climate
and Earth system modelling) and the IPCC over the past 4 decades.

With a new IPCC AR cycle (AR7) beginning, it is timely to review how the international modelling community has
supported climate policy in the past, including earlier AR cycles, and ask what advances can be made in the overall quality
and availability of science to support policy needs. In addition, it is pertinent to review our current understanding of, and
ability to model, coupled Earth system change, as well as the societal and environmental impacts associated with this change
and ask whether plausible, safe pathways can be developed for the Earth system that avoid the worst impacts of this change.
Many of the international projects listed in Table 1, that provide the scientific knowledge on which IPCC reports are based,
are beginning new cycles. For example, CMIP7 is starting to take shape, likely running through to ~2030 or beyond. In this
paper we outline a number of areas we believe the international modelling community can significantly advance our
understanding of, and ability to simulate, past and future Earth system change, including the impacts of these changes.
Progress in the proposed areas will also allow an improved investigation of mitigation options for limiting long-term global
warming, and its impacts, to acceptable levels. Such developments will deliver enhanced scientific support to international
climate policy, during and beyond AR7. The advances we propose assume continued *development*, *expansion, maintenance,*
and *integration* of a robust and interconnected infrastructure ecosystem. Such an infrastructure has underpinned past
international modelling collaborations and is a fundamental requirement for realizing the ambitious goals outlined here. The
specific science, and science for policy, ambitions, as well as the necessary underpinning infrastructure, are discussed in
more detail in subsequent sections. Each proposed focus area can be summarized by the following key goals:

● **Provision of a coordinated, internally consistent set of simulations, data, and knowledge to support IPCC**
**assessments and international climate policy.** The resulting data sets and knowledge should be based on the most
recent and consistent set of Integrated Assessment Model (IAM) scenarios, global and regional Earth system model

(ESM) projections and simulated societal and environmental impacts. With consideration of impacts arising both due to the projected Earth system change, and directly from any mitigation actions assumed in the IAM scenarios.

- **Improving understanding and guidance on future Earth system change, allowable emissions, net-zero responses, and safe, long-term pathways for planet Earth.** Ensure global and regional ESMs, IAMs, and impact models include the required level of process realism, process interactions, and consistent forcing data to accurately simulate the response of the Earth system and human societies to future socio-economic, mitigation, emission, and land-use scenarios. Develop and analyse a range of future pathways that limit long-term global warming to less than 1.5 or 2°C above pre-industrial levels, while minimizing the negative impacts on society and the environment.

- **Improving our understanding of, and ability to simulate key climate processes, climate variability, extreme events and regional impacts.** Ensure global and regional climate models (GCMs and RCMs) accurately represent key processes, couplings, modes of variability and feedbacks that underpin global to regional climate change. Use these models to deliver robust and detailed projections of regional climate change, including changes in extreme events. Ensure the socio-economic information used to develop IAM mitigation and scenario data is suitably disaggregated and combined with climate projection data to support national to regional scale impact assessment, adaptation planning and climate services.

- **Increasing collaboration across approaches to further improve global and regional Earth system and climate models.** Ensure strong collaboration across efforts to; increase process realism and coupling in ESMs, increase model resolution and improve physical parameterizations, including ML hybrid-modelling approaches. Ensure these approaches are optimally combined to deliver the best possible development pathway for the next generation of Earth system models.

- **Improving model simulations of the observational record and key metrics of climate change.** Ensure improvement in the simulation and understanding of the observed, historical evolution of climate, particularly historical global and regional warming, encompassing the forcings, processes, and feedbacks that determine the rate and pattern of this warming. Improve our ability to constrain and simulate key climate change metrics, such as the Effective Climate Sensitivity (EffCS), Transient Climate Response (TCR), the Transient Climate Response to cumulative carbon Emissions (TCRE) and the Regional Warming to Global Warming ratio (RW/GW)

- **Sampling and quantifying future uncertainty.** Develop and apply a hierarchy of models and methods to efficiently explore the range of uncertainty inherent in future Earth system change and its impacts. Ensure regional and national scale adaptation and mitigation is informed by a more complete sampling of the range of potential climate futures, including rare (high impact, low likelihood) outcomes, their local climate signature, and the potential consequences of these for society, the environment and climate policy.

- **The underpinning technological infrastructure.** Further develop and maintain a robust, globally inter-connected infrastructure ecosystem to ensure efficient co-production and co-exploitation of internally consistent model simulations, via information, data and computational services that enable the rapid and reliable sharing of requirements, knowledge, data, and analysis tools. Such sharing needs to be both within and across multiple modelling projects and user communities, as well as providing suitable support to policymakers, planners, climate services, and the wider international research community.

| Acronym | Initiative or project name | Website | Main themes | Citation |
|---|---|---|---|---|
| IAMC | Integrated Assessment Modelling Consortium | https://www.iamconsortium.org | Future socio-economic pathways, emission and land use scenarios | Moss et al., 2010 |
| WCRP CMIP | Coupled Model Intercomparison Project | https://wcrp-cmip.org/ | Earth system and Global Climate modelling | Eyring et al., 2016 |
| ScenarioMIP | ScenarioMIP | https://wcrp-cmip.org/model-intercomparison-projects-mips/scenariomip/ | Further develop IAM scenarios into emission, concentration and land-use scenarios for CMIP and CORDEX. | O'Neill et al., 2016 |
| WCRP CORDEX | Coordinated Regional Downscaling Experiment | https://cordex.org | Regional climate downscaling | Giorgi et al., 2009 |
| VIACS AB | Vulnerability, Impacts, Adaptation & Climate Services Advisory Board | https://viacsab.gerics.de/ | Advisory body for linking CMIP and CORDEX to the impacts and climate services communities | Ruane et al., 2016 |
| ISIMIP | Inter-Sectoral Impact Model Intercomparison Project | https://www.isimip.org | Global and regional impact modelling for multiple sectors | Frieler et al., 2017 |
| ESGF | Earth System Grid Federation | https://esgf.llnl.gov/ | Data curation and distribution system for CMIP and CORDEX | Balaji et al., 2018 |

**Table 1. Examples of the main international projects contributing to the provision of simulations, data and scientific knowledge to**
**support climate policy, particularly IPCC assessment reports, including a main reference for each activity. CMIP and CORDEX**
**are coordinated by the World Climate Research Program.**
Over the past few years a number of papers offer important perspectives on future priorities for Earth system and climate
modelling, focussing on; the benefits of increased model resolution (Satoh et al., 2019; Palmer and Stevens, 2019; Slingo et
al., 2022), the role of AI and ML in model development (Bauer et al., 2023; Eyring et al., 2024b; Schneider et al., 2024),
development of Digital Twins (Bauer et al., 2021; Hoffman et al., 2023; Bauer et al., 2024), priority areas for CMIP7 (Dunne
et al., 2023; Sanderson et al., 2023), proposals for an operational approach to CMIP (Jakob et al., 2023; Stevens, 2024), and
future scenarios to support the IPCC process (Pirani et al., 2024). The recommendations we present here should be viewed in
the light of these papers and summarize the views of a group of European scientists who have been engaged in, and in a number
of cases led, major international modelling exercises that have delivered critical support to past IPCC assessment cycles. A
similar perspective piece, from a number of U.S. climate modelling centres, has also recently been published (Mariotti et al.,
2024). Our perspective aims to address the range of activities involved in delivering actionable scientific support to
international and national climate policy and therefore encompasses; IAM-based socio-economic, emission and land use
scenarios, global and regional Earth system and climate models, regional downscaling and calibration, projection ensembles
and emulators, uncertainty quantification, sectoral and environmental impact models, as well as the computational
infrastructure necessary to realise and disseminate this complex workflow.
207 .

## 2      Provision of a coordinated, internally consistent set of simulations, data, and knowledge to support IPCC assessments and international climate policy.

The process by which the aforementioned activities have, in the past, delivered data and knowledge into the science and policy arenas is summarized in Fig. 1. IAMs develop a range of future global pathways, based on narratives for socio-economic, political, and technological development, as well as climate policy. For methodological reasons these scenarios do not (yet) consider the impacts of future climate change on human behaviour. The pathways are typically quantified in terms of highly aggregated information on future population and economic development, energy and food system development, and environmental consequences. For each pathway, marker anthropogenic emission and land-use scenarios are selected (van Vuuren et al., 2011; O'Neill et al., 2016; Riahi et al., 2017). These scenarios are combined with observation-based estimates for the historical past, resulting in a time series of emission and land use data covering ~1850 to 2100 (Hurtt et al., 2011; Gidden et al., 2019). Using simple climate models (e.g. MAGICC; Meinshausen et al., 2011) and chemistry-climate models (Lamarque et al., 2011), the emissions are converted into atmospheric concentration time series. The concentration timeseries, along with the land-use scenarios, are used to "force" ESMs in CMIP to investigate potential changes in the Earth system arising from each scenario. The ESMs deliver time-varying, spatially discrete estimates of the past and future evolution of the Earth system, sampling the range of available emission and/or concentration scenarios (Tebaldi et al., 2021). CMIP simulations are extensively used to inform policymaking addressing global climate change risks. They are also made available to the international research community via the ESGF, where they are used to increase understanding of the Earth system and Earth system change, and to highlight areas requiring further model improvement.

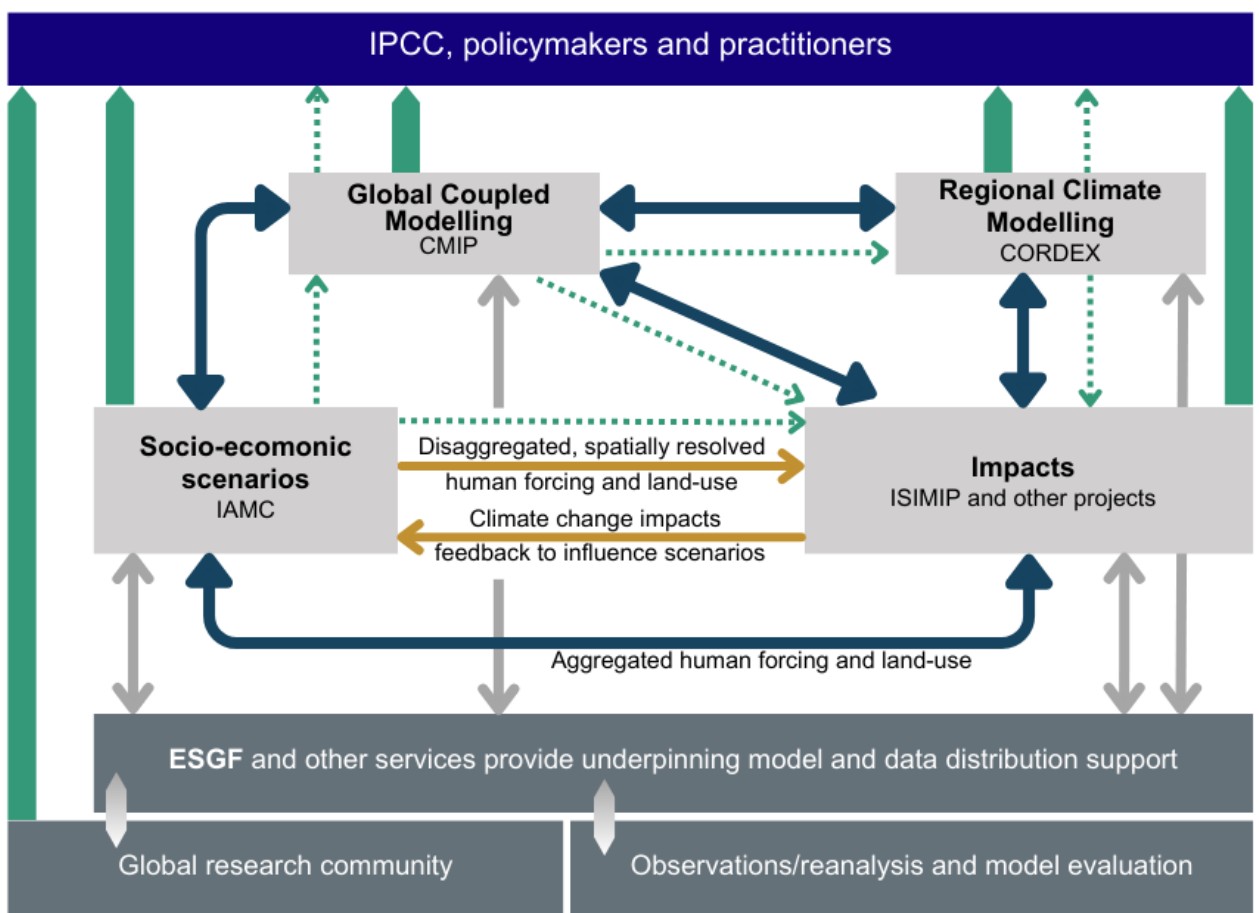

**Figure 1: A schematic illustration of how earlier rounds of IAMC, CMIP, CORDEX and impact modelling activities, such as ISIMIP, have worked together to develop and apply future socio-economic and emission scenarios (IAMC), increase the scientific understanding of, and ability to simulate the coupled Earth system (CMIP and CORDEX), and investigated the impacts of Earth system change on societies and the natural environment (ISIMIP etc). In the figure dark blue lines illustrate the main (generally**

**two-way) exchanges of scientific knowledge between the different projects. Dotted green lines indicate the main (simulation) data**
**transfer between projects, while grey lines show the main data exchanges outside of these projects (e.g. onto the ESGF for open use**
**by the global research community or into regional or national data distribution sites). Thin orange lines illustrate the new exchanges**
**proposed in Sect. 2 of this paper. Finally, the thick green lines illustrate the main knowledge and data exchange routes between the**
**different projects, the global research community, and the IPCC assessment process, as well as with multiple policymakers,**
**practitioners, and climate service providers around the world.**

CMIP simulations are used extensively as boundary forcing for regional downscaling (e.g. CORDEX) to generate climate
information at spatial scales relevant for adaptation policy and climate services, as well as to drive impact model simulations
(e.g. crop models in AgMIP (Ruane et al., 2017), fisheries and marine ecosystem models in FishMIP (Tittensor et al., 2018),
and a range of impact models that contribute coordinated simulations to ISIMIP (Frieler et al., 2017), addressing impacts
such as, biome changes, water resources, human health, energy systems and biodiversity). Regional downscaling follows two
main pathways; (i) dynamical downscaling generates high-resolution regional simulations consistent with the ESM boundary
condition data (Ruti et al., 2016; Jacob et al., 2020; Teichmann et al., 2021) and (ii) empirical-statistical downscaling
(including ML methods) combine observations and models to translate large-scale features simulated by the ESMs to high-
resolution, local scale climate information (Gutiérrez et al., 2018; Lange, 2019; Karger et al., 2023). Impact models use both
CMIP and CORDEX climate data, as well as socio-economic data and information on mitigation actions from the IAM
scenarios (e.g. population distributions and land use patterns that include information on mitigation measures), as forcing to
assess the societal and environmental impacts arising from the range of simulated futures (Frieler et al., 2017).
The combined outcome of this international effort are a set of simulations, data and resulting knowledge covering the past
~175 and future ~100 years (and sometimes longer) that sample; (i) plausible future global socio-economic development
pathways, (ii) emission, concentration and land-use scenarios commensurate with these pathways, (iii) global and regional
Earth system changes associated with each future pathway and (iv) the societal and environmental impacts arising from the
simulated Earth system changes, as well as direct impacts associated with the socio-economic and/or mitigation measures
applied in the IAM scenarios.
There are numerous challenges involved in running the number and variety of model simulations across this range of
activities, including cross-project and cross-model dependencies. As a consequence, to date it has not been possible to
develop a single, coordinated dataset of forcings, simulations and findings from all four activities (IAMs, CMIP, CORDEX,
impact modelling), based on a common set of socio-economic assumptions, scenarios, and driving data, within a single IPCC
Assessment cycle. This limitation reduces the overall consistency and utility of information entering the three IPCC working
groups (WGs). For example, Global (CMIP) and Regional (CORDEX) simulations are often out of sync, with CORDEX
RCMs using boundary data derived from an earlier phase of CMIP. A similar example holds for impact models that often
use a mix of global and regional forcing from different phases of CMIP and CORDEX. Furthermore, impact models forced
by CMIP/CORDEX climate data, do not include all the socio-economic and climate policy information that underpin the
driving IAM emission and land-use scenarios. This is particularly acute with respect to a number of direct human forcings.
These forcings are aggregated across multiple sectors and large spatial scales in the IAM scenarios, but need to be
disaggregated and harmonized with observed historical data, to more detailed spatial scales and individual sectors, to allow
an accurate estimate of their impact on society and the environment, in combination with the impacts due to Earth system
change (e.g. see *Direct Human Forcings*, as listed on Table 1, Frieler et al., 2024). An improved accounting of such direct
human forcings will be increasingly important as future scenario pathways include major (human) interventions likely
required to deliver the negative $CO_2$ emissions necessary to achieve the Paris Agreement targets. Such interventions
themselves can have important direct impacts on food production and biodiversity and therefore need to be accounted for in
impact assessments.

Partly for methodological reasons, the impacts of climate change (and the potential societal responses to these changes) have not been included in IAM scenarios describing future socio-economic trajectories (i.e. Shared Socio-economic Pathways (SSPs), O'Neill et al., 2020). As climate change is expected to have a considerable impact on society, it is important methods are developed that allow these feedbacks to be included in future scenario development (Pirani et al., 2024). Ideally information on the impacts of climate change would be fed back into the IAMs to iteratively generate new future socio-economic and policy pathways that include the societal responses to both the applied climate mitigation measures and to the impacts of climate change. For example, future land use will need to be adjusted to satisfy global food production, while accounting for the impacts of climate change on crop yields and changes in available land resulting from any land-based climate mitigation measures. These iterative adjustments to future socio-economic scenarios are one way to represent societal adaptation to projected climate change. Given the tight timelines it will not be possible to fully develop such iterative and interactive steps within the IPCC AR7 cycle. Nevertheless, we recommend urgently addressing this link as the envisioned modification of workflows has the potential to significantly improve the overall consistency of future scenarios, integrating important information across socio-economic, Earth system and impact projections.

The lack of consistency, of both data and knowledge entering IPCC and national climate change assessments, reduces its overall utility and makes the interpretation of uncertainties across the various data sources a challenge. This can lead to inconsistent data and knowledge being used to develop climate policy, with some data being more than 10 years old. We believe the time is right to much more tightly link these key international activities, with more extensive and rapid sharing of simulations, data, knowledge, tools, and personnel, moving such critical *science for policy* work towards an operational footing. Such a change has been proposed earlier (e.g. Jakob et al., 2023; Stevens, 2024). We agree with these proposals but stress the need for *"operationalization"* across the entire workflow involved in developing and delivering robust and useable scientific knowledge. This includes; generation of IAM scenarios and associated forcing data, global and regional Earth system model simulations based on these scenarios, impact model simulations, post-simulation evaluation and analysis, uncertainty quantification, science to policy knowledge translation, and the technical infrastructure needed to support the entire endeavour. To maximize the relevance and utility of the resulting science for policy, we further propose such operational activities employ a co-development and co-exploitation approach, where a cross-section of intended users of the science are involved throughout the process.

Such developments require support across a number of international coordinating bodies, as well as mechanisms to coordinate or pool the significant funding required, for what is inherently an international, multi-institutional and multi-disciplinary endeavour. The building blocks for this do exist, represented by IAMC, CMIP, CORDEX, VIACS, ISIMIP and the ESGF. To date, the bulk of the effort to realize these interconnected projects have been funded through short-term, competitive research grants, with the availability and international coordination of this funding arising partly by chance and often thanks to shared IPCC timelines (Meehl, 2023). While such a development requires significant effort, funding and coordination, the long-term benefits for climate policy are potentially very significant. While moving the policy- and service-oriented aspects of climate projections and impact assessment towards a more operational approach is important, we stress the paramount importance of maintaining a strong science understanding, model improvement, and open data access, approach across all these activities. This will help maintain global participation and ensure continual improvement in the quality of data and knowledge entering the climate policy and service arenas. Fully achieving these goals on the timescale of IPCC AR7 will not be possible. Nevertheless, a first step in this direction is under development as part of the planning for CMIP7, which will operate a dual timescale approach. A set of CMIP7 Fast Track (FT) simulations, specifically intended to support IPCC AR7, is under development. The CMIP7 FT aims for a small set of policy relevant experiments that can be

rapidly performed and made available for analysis by early 2027. In addition to the Fast Track, the bulk of CMIP7 will operate on a slower timescale, roughly from 2025 to 2030, with individual science-oriented MIPs (Model Intercomparison Projects) developing and realising a range of experiments and analyses to address outstanding questions and challenges in Earth system modelling.

Starting to develop a more joined up and efficient workflow across projects, along with increased internal consistency of data and knowledge emanating from these projects, will be an important step towards a durable, more operational approach to delivering scientific support to climate policy and climate services.

**3    Improving knowledge and guidance on future Earth system change, allowable emissions, net-zero responses, and safe landing pathways for planet Earth.**

**3.1    The Paris Agreement: The risk of warming overshoot, allowable emissions, net-zero and negative emissions, and Earth system feedbacks.**

The 2015 Paris Agreement (with an aim to limit long-term global warming to well below 2°C above pre-industrial temperatures and pursue efforts to limit warming to 1.5°C; Riahi et al., 2021) focused the attention of policymakers and the public onto the risks and consequences of exceeding these key targets. Partly in response to such policy needs, work accelerated on quantifying allowable carbon emission budgets commensurate with the Paris goals (Millar et al., 2017; Rogelj et al.,2019; Lamboll et al., 2023). It became increasingly clear that to provide accurate guidance on such allowable budgets, Earth system models needed to improve their representation of the carbon cycle and its interaction with physical climate processes. In addition, further improvement was required in representing non-$CO_2$ climate forcers, such as methane, nitrous oxide and aerosols. Focus also turned to the risk of triggering feedbacks that might push temperatures further from a given target, once the target was exceeded, as well as on the risk of exceeding Earth system tipping points, with potentially major regional impacts. Lastly, recognition that international policy would likely lead to the climate being stabilized at temperatures warmer than pre-industrial or present-day, stimulated work to better quantify the long-term consequences associated with such a stabilized warmer world (King et al., 2021).

Over the past decade significant progress has led to several ESMs now including a full representation of the carbon cycle, interactively coupled to the physical climate (Arora et al., 2020). This progress has motivated calls for CMIP7 to more strongly focus on $CO_2$-emission driven simulations, where a more complete representation of future climate – carbon cycle feedbacks can occur (Sanderson et al., 2023). A number of ESMs are also incorporating and coupling other Earth system processes required to properly investigate future emission pathways that realise the Paris Targets, as well as the consequences of long-term stabilization. Developments include; nutrient limitation on terrestrial carbon uptake (Lawrence et al., 2019; Wiltshire et al., 2021), interactive methane cycles with the ability to run in emission-mode for methane (Folberth et al., 2022), interactive treatment of nitrogen and iron cycles (Dunne et al., 2020), interactive permafrost (Burke et al., 2020, Schädel et al., 2024), interactive fires (Mezuman et al., 2020; Teixeira et al., 2021), full atmosphere chemistry (Gettelman et al., 2019; Archibald et al., 2020) coupled to advanced aerosol models (Mulcahy et al., 2020), as well as interactive Greenland and Antarctic ice sheets (Smith et al., 2021; Muntjewerf et al., 2021). Many of these developments, occurring across several ESMs, have either recently entered use in coupled model configurations, or are in an advanced stage of development and planned for use in CMIP7. As a result, the Earth system modelling community are entering a period where simulation of the full Earth system during overshoot, recovery, and long-term stabilization can deliver critical new insights that are urgently required to inform international climate policy.

An important focus for CMIP7 and ScenarioMIP (O'Neill et al., 2016; van Vuuren et al., 2023) therefore, will be investigation of plausible emission scenarios and global warming pathways that successfully realize the Paris Agreement. Key questions within this activity include; What is the feasibility of actually realizing the Paris targets? Whether a temporary warming overshoot is inevitable? If so, what rate and magnitude of warming is likely to occur, and how sensitive is the Earth system to such factors? Additionally, is it feasible to return to a target warming level on a reasonable timescale once an overshoot has occurred (Bauer et al., 2023)? To provide robust policy guidance on the plausibility and consequences of such pathways, several additional questions need to be addressed: Can accurate predictions of carbon emission budgets (and budgets of other radiatively important greenhouse gases) be made that are commensurate with different warming targets, with or without overshoot (Ramboll et al., 2023)? What is the role of anthropogenic aerosol emissions with respect to future warming and achievability of the Paris targets (Jenkins et al., 2022; Wang et al., 2023) What is the risk of amplifying feedbacks being triggered during overshoot (Melnikova et al., 2022), and is there a risk of exceeding tipping point thresholds in the Earth system, society or the natural environment, during overshoot (Wunderling et al., 2023)? If plausible negative emission pathways do exist, that return the Earth system to an acceptable temperature at an acceptable rate, once overshoot has occurred, what will be the environmental consequences of following these pathways? Furthermore, during the overshoot phase, if major changes or impacts (e.g. ecosystem degradation, population displacement, economic damages) do occur, or tipping points are exceeded (either in society or the Earth system), are these changes reversible when temperatures return back below a target level (Kim et al., 2022; Reed et al., 2023; Santana-Falcón et al., 2023) and how long will such a recovery take (Albrich et al., 2020, Meier et al., 2012)?

Existing mitigation pathways that rely on negative $CO_2$ emissions assume a significant stimulation of terrestrial carbon uptake through extensive modifications to land-use (Smith et al., 2016). How the carbon cycle will respond to these interventions is not well quantified. Nor is the actual efficacy of these interventions in reducing temperatures (Schleussner et al., 2023), or the ensuing impacts on the natural world, particularly biodiversity. A dominant part of the negative $CO_2$ emissions in present IAM scenarios is assumed to come from the AFOLU (agriculture, forestry and other land use) sector, through large scale deployment of bioenergy with carbon capture and storage (BECCS). It is of the utmost importance ESMs, with a comprehensive process-based representation of the carbon cycle, are used to assess the efficacy of such AFOLU scenarios in terms of realized negative emissions and temperature response, accounting for interactions with the natural carbon cycle and regional climate. Such major changes to the land surface will likely also lead to significant impacts on water availability, biodiversity and a range of human activities (Séférian et al., 2018; Hof et al., 2018), both directly from the change in land use and indirectly through induced changes in regional climates. Such potential impacts need to be carefully assessed with impact models, with any negative impacts contrasted against the positive impact of the mitigation actions on global warming. New negative $CO_2$ emissions technologies that encompass marine-based $CO_2$ removal (mCDR) are increasing in interest. Such approaches aim to increase marine carbon uptake through ocean alkalinization (Kwiatowski et al., 2023; Palmieri and Yool, 2024) or increase the storage of ocean carbon via marine afforestation (Bach et al., 2021). These new approaches have the potential to reduce the demand on land-based CDR, reducing the impacts of these techniques on land. However, such ocean techniques can lead to negative consequences for marine ecosystems and organisms, by altering marine nutrients cycles. It is important to emphasise that the full Earth system response to marine CDR is as uncertain as its land counterpart. Uncertainties in its efficacy to remove and store $CO_2$ remain poorly quantified and estimating the lifetime of $CO_2$ storage in the water column represents an additional challenge compared to the land-based CDR, due to the complicating role of ocean circulation and potential redistribution of $CO_2$.

In addition to negative $CO_2$ emissions, Solar Radiation Management (SRM) has been proposed as an alternative (or additional) route to limiting global warming to 1.5°C. While there remain concerns around the unintended consequences of

SRM (Bonou et al., 2023), as well as the long-term governance of such technology (Pasztor and Harrison, 2021), the international SRM community recently designed a set of scenarios that allow investigation of both the efficacy and potential climate impacts of such technology (MacMartin et al., 2022; Baur et al., 2023; Baur et al., 2024). The same community have proposed an experiment protocol for the CMIP7 Fast Track (Visioni et al., 2024) that targets recovery of the global mean surface temperature to 1.5°C threshold after overshoot. As the world continues to get closer to the 1.5°C threshold, interest in SRM and geoengineering more broadly is likely to increase. The science community will be asked to provide the best possible guidance on the efficacy of SRM, the potential climatic and ecological impacts of SRM, as well as information on the scales (temporal, spatial and quantity) required for this technology to deliver long-term, safe climate stabilization. Such work on climate 'solutions' including SRM should be organized under the WCRP Lighthouse Activity on Climate Intervention, which brings together international research communities focussing on both CDR and SRM.

Finally, once an "acceptable" warming level is reached, it remains to be established whether the Earth system can be stabilized, long-term at this level (Jones et al., 2019)? And, if so, what the consequences across the Earth system and for society will be from such stabilization (King et al., 2021; Palazzo Corner et al., 2023)? All these questions have major implications for international climate policy. Reliable answers are urgently needed. The international research community is beginning to address such questions, and increasingly has the tools capable of providing answers. We believe the new round of international modelling projects have the potential to make major advances towards delivering robust answers.

Past CMIP cycles, including the most recent phase CMIP6 (Eyring et al., 2016a), emphasized $CO_2$-concentration driven simulations, where atmospheric $CO_2$ concentrations are prescribed and simulated carbon cycle – climate feedbacks cannot influence atmospheric $CO_2$. This approach was taken largely for pragmatic and inclusivity reasons (i.e. there was only a relatively small number of models with robust and stable coupled climate and carbon cycles). Thanks to efforts such as $C^4MIP$ (Friedlingstein et al., 2006, Arora et al., 2020), this is no longer the case, with a significant number of ESMs now including advanced carbon cycles coupled to their physical climate (Sanderson et al., 2023). Due to the small remaining carbon budgets involved in realizing the Paris targets, and uncertainty in how the carbon cycle will respond to negative and net zero emissions, it is imperative more ESMs in CMIP7 run in $CO_2$-emission mode, with full interaction between the physical climate and carbon cycle, including prognostic atmospheric $CO_2$ (Sanderson et al., 2023; Gier et al., 2024). This will support an improved assessment of feedbacks involving the physical climate and the carbon cycle, addressing consequences for allowable future carbon emissions, the amount of negative emissions required after different overshoot to achieve different stabilization goals, and the associated risks, impacts and potential for irreversible change across the Earth system. Only through such a coupled, prognostic approach can anthropogenic $CO_2$ emission scenarios, intended to realize key warming targets, be connected with the Earth system response and the impact of these responses on atmospheric $CO_2$ and realized warming/cooling pathways.

We propose other important aspects of the coupled Earth system, at risk of rapid change, should also be run in a more *coupled and prognostic* manner in CMIP7. Assessment of coupled interactions and risks across the entire Earth system, including potential tipping point risks (Ritchie et al., 2021), is severely lacking in earlier IPCC Assessment Reports. Giving greater emphasis to coupled and prognostic interactions across the Earth system (particularly those thought to play a major role in determining the magnitude of future change) in an internally consistent framework will allow a more complete assessment of Earth system change, beyond that focussed solely on the physical climate. In addition, we emphasize the need to assess the impact of specific and targeted human actions (designed to mitigate future climate change or to adapt to expected future change) on regional climate, as well as on other aspects of the coupled Earth system, including resilience of the natural environment, biodiversity, and consequences for other human activities (e.g. food security, energy production or

air quality).  The current scientific priorities with respect to such interactions, along with (in italics) the key phenomena, feedbacks and consequences such coupled simulation would enable improved assessment of, are listed below:

(i)     Water, vegetation and biogeochemical cycles of carbon, nitrogen, phosphorous; *improved estimates of vegetation change, terrestrial carbon uptake, regional water cycles and ecosystem tipping risks.*

(ii)    Climate, vegetation, and fire: *improved assessment of future fire risk and interactions with carbon uptake, atmospheric composition and ecosystem tipping risks.*

(iii)   Permafrost, climate, vegetation, and carbon: *stability of permafrost under warming and long-term warming stabilization, carbon/methane release from thawing permafrost, ecosystem expansion into thawing permafrost zones.*

(iv)    Climate, ice sheets, and sea level: *improved assessment of potentially irreversible loss of Antarctic and Greenland ice mass and consequences for sea level rise, ocean circulation and ocean heat uptake.*

(v)     Climate, atmospheric composition, and air quality: *internally consistent assessment of regional radiative forcing, climate change and air quality.*

(vi)    Ocean physics, biogeochemistry and ecosystems: *assessment of ocean warming, marine carbon uptake and long-term storage, ocean acidification and impacts on marine ecosystems.*

(vii)   Human-Earth System interaction: *assessment of the direct impact of human activities on the Earth system, regional climate, society, and the environment. e.g. Mitigation actions designed to address air quality and/or climate change, such as major land use change, nature-based solutions, climate interventions (geoengineering). Adaptation measures designed to address regional to national scale climate risk.*

(vii)   The interplay between global change, regional climate variability, changes in climate and weather extremes, and resulting impacts across the Earth system.

**3.2     Regional Earth system change; assessing societal and environmental impacts.**

In addition to changing how global ESMs are run, we propose that regional downscaling (for example dynamical downscaling or Regional Climate Modelling, as used in CORDEX) also advance their representation of key regional Earth system processes (beyond the physical atmosphere-land system; Giorgi and Prein, 2022; Nabat et al., 2020; Sevault et al., 2014). Here we refer to regional climate modelling or dynamical downscaling in the broadest sense, encompassing any physics-based dynamical model targeting a fine-scale representation of the climate over a specific region of the world. This includes limited-area models (LAM), variable-resolution GCMs (VRGCM) and, more recently, regional earth system models, convection-permitting regional models, and two-way coupled systems. In addition, atmosphere-land only global models are beginning to run for decadal timescales (and likely longer in the coming decade) and can be driven by sea surface temperatures and sea ice derived from ESM projections, providing a global downscaling option for coupled ESM projections. Whatever the technical choices used to perform such dynamical downscaling in future projection mode, forcings from global ESMs and GCMs will be required, either as lateral, surface, or inner model boundary condition data. Similarly, we use the term statistical downscaling in a very broad sense, covering established statistical methods for transferring

simulated large-scale climate data to local scales, as well as the increasing range of machine learning (ML) techniques, including recent deep learning applications (Gerges et al., 2023; Soares et al., 2024).

To better sample the uncertainty range of global projections, dynamical and statistical downscaling should preferentially use $CO_2$ emission-driven ESMs as boundary forcing and employ an efficient (as automated as possible) method to select an ESM ensemble for a given region and rapidly generate the required boundary condition data. The resulting combination of global emission-driven ESMs, regional ESMs, and advanced statistical/ML-based downscaling, running in a tightly linked framework, will allow a more complete assessment of potential changes across the global and regional environment at scales required by policymakers and planners. Given the rapid development of a diversity of dynamical, statistical and ML-based methods to generate high-resolution regional data, it is important a common evaluation framework is developed that is applicable across global to local scales (and across the implied model resolutions) as well as being agnostic to the methods employed, so different downscaling approaches can be objectively evaluated against each other, region by region and application by application.

We further recommend impact models use a coordinated, multi-model ensemble of (global and regional) simulation-data, based on the CMIP7 $CO_2$-emission driven ESMs, that capture a representative fraction of the uncertainty space of global and regional projections. In addition, impact models should aim to sample multiple members of individual ESMs, and the downscaling of these ESMs, to better quantify the importance of internal (natural) variability in regional climate impacts. Forcing impact models, either directly by global ESM output or by appropriately downscaled data, themselves driven by the same ESM simulations, will ensure global consistency of the impact simulations and comparability of impacts resulting from global and regionally downscaled forcing over the same region. In addition to coordinated forcing from ESM and downscaled data, a more complete, disaggregated set of IAM scenario data describing socio-economic development and potential mitigation or adaptation measures will ensure greater coherency between global and regional impact assessments and the underpinning IAM, ESM and regional forcing data. The resulting global models and downscaling combinations can also be used to assess the efficacy and potential impacts associated with different regional climate change mitigation or adaptation actions, offering scientific assessment of such proposed climate solutions.

## 4   Improving our understanding of, and ability to model key climate processes, climate variability, extreme events and regional impacts.

### 4.1   Improving key phenomena and couplings in global climate models.

Some of the key uncertainties in Earth system model projections relate to errors in simulating important regional climate processes and phenomena, including interactions across spatial scales and regions. For some of these phenomena, model resolution has been shown to be a key factor. Hewitt et al. (2022) showed that increasing ocean model resolution, in particular better resolving the ocean mesoscale, is important for accurately representing a number of key processes, including; ocean eddies in the Southern Ocean and North Atlantic (*with implications for simulated marine heat and carbon uptake, ice sheets and sea-level rise*), ocean deep water formation in the Labrador and Nordic Seas and on the Antarctic shelf (*with implications for the global ocean overturning circulation and heat uptake*), the Atlantic Meridional Overturning Circulation (*with implications for heat and carbon uptake, as well as regional climate*), ocean upwelling regions (*with implications for marine carbon uptake, productivity and fisheries*). Increased resolution, in both the atmosphere and ocean, is also important for simulating large-scale hydrological processes (Vannière et al., 2019) (*with important implications for regional water cycles, water availability and food security*), as well as modes of climate variability, such as the El Niño Southern Oscillation (ENSO) and associated teleconnections (*with implications for the rate of ocean heat uptake and*

regional climate variability). While increased model resolution (to better resolve the meso- or the synoptic scales) is an important component of reducing several systematic biases in coupled models, it is equally important to improve key parameterization schemes for processes that continue to be unresolved, even at horizontal resolutions of ~10km/0.1° in coupled models. In particular, it is critical to ensure further improvement in parameterizations at the heart of uncertainty in the simulated Effective Climate Sensitivity (EffCS), Transient Climate Response (TCR) (Meehl et al., 2020) and aerosol-cloud forcing (see Sect. 6 of this paper).

Upscale effects from many small-scale processes can be important. For example, oceanic mesoscale eddies tend to drive atmospheric mesoscale storms in the extra tropics (Liu et al., 2021), while at larger scales the atmosphere can drive ocean variability (Frankignoul, 1985). These effects are apparent only in coupled systems and their large-scale consequences, such as the preferred location and orientation of the jet stream, mid-latitude storm tracks, and related air-sea fluxes, can only be captured in large-domain models with mesoscale or better resolution (Seo et al., 2023). Furthermore, couplings between the heat, water, and carbon cycles, means improving the representation (and parameterization) of physical processes will deliver important benefits for simulating the carbon, and other biogeochemical, cycles. In addition to the large-scale impacts, higher resolution models also offer an improved simulation of climate variability, in particular weather extremes such as; tropical cyclones (Roberts et al., 2020), extreme precipitation (You et al., 2023), atmospheric rivers (Liang and Yangyang, 2023), jet streams and atmospheric blocking (Schiemann et al., 2020) with consequences for the frequency and location of extreme weather (Athanasiadis et al., 2022), which both depend on SST realism delivered by resolving the ocean mesoscale. All these events have important impacts across the coupled Earth system, including upscale effects, e.g. drying of the atmospheric column by tropical cyclones over the Maritime Continent, with impacts on ENSO (Scoccimarro et al., 2021). Similarly, in the ocean increased resolution can improve the representation of important dynamical phenomena, such as marine heatwaves (Plecha and Soares, 2020) the representation of bottom water formation (Heuzé, 2021) and mixed layer eddies (Calvert et al., 2020).

Increasing model resolution alone does not guarantee improvement in all simulated metrics and leads to significant challenges related to model spin-up, model equilibration, calibration, and uncertainty quantification. Simulation improvements are often best realized through a combination of increased model resolution and targeted improvement to key parameterization schemes. While the compute cost increases considerably as model resolution is increased, recent studies suggest increased resolution can deliver important insights into some long-standing model biases, and perhaps reconcile mismatches between simulated and observed historic trends. For example, Rackow et al. (2022) show that resolving the ocean mesoscale improves the simulation of Antarctic sea-ice trends, Chang et al. (2023) illustrate increased realism in ocean upwelling as model resolution is increased, and ongoing work suggests higher resolution simulations can better capture recent observed trends in the Eastern Pacific that are not captured in CMIP6 models (Seager et al., 2022). Such improvements will increase confidence in future model projections and have important implications for predicting future extreme events, such as tropical cyclones, floods, droughts, and heatwaves.

There is strong evidence a coordinated set of simulations for CMIP7, with resolutions enhanced over those typically used (e.g. 10-20 km in the atmosphere and ~0.1° in the ocean), can deliver an improved simulation and understanding of key regional climate processes and a more robust assessment of future changes in many of these processes, with benefits for impact and adaptation planning. Chang et al. (2020) demonstrated that CMIP-length simulations, with an equilibrated coupled model, are now possible at resolutions of ~10-20 km/0.1°. Many groups produced simulations following the CMIP6 HighResMIP protocol (Haarsma et al., 2016), though generally with very limited ensemble sizes. Given increased model efficiency and available compute resources, CMIP7 provides an opportunity to further investigate the benefits of increased

coupled model resolution, alongside increased ensemble size, longer simulation length, methods for improved model equilibration and initialization, and enhanced process realism. Given current structural limitations of coupled climate models, of whatever resolution, sampling model diversity, through multi-model CMIP-style exercises, remains critical for providing robust estimates of projection uncertainties and risks (see Section 7). This is particularly the case with respect to regional climate change, where processes may be resolution-dependent (e.g. Moreno-Chamarro et al., 2022) and therefore sensitive to biases common across lower resolution models. A diversity of enhanced resolution coupled models thus needs to be promoted, but also optimized across the competing demands for delivering future projection data that is of maximum quality and utility both for the science and policy communities.

## 4.2 Increased model resolution from global to regional scales for regional impact assessment and adaptation.

Like their global counterparts, Regional Climate Models have also increased in resolution, with a growing set of models now running at convection-permitting resolutions (~1-3km resolution; Ban et al., 2021; Hohenegger et al., 2023). In addition to an improved simulation of the convective scale, high-resolution itself brings direct benefits, by delivering climate information closer to impact and adaptation relevant scales and by better resolving local climate in regions of strong orographic forcing, complex land-sea-lake structures, or heterogeneous land surface types. Moreover, explicitly resolving convective events, including the self-organization and self-intensification of these events, brings physical grounding to simulated precipitation extremes (Kendon et al., 2021; Caillaud et al., 2024), including the ability to evaluate models against observations at common spatial scales (Caillaud et al., 2021). A growing set of regional projections, employing convection-resolving models (Pichelli et al., 2021; Chapman et al., 2022; Kawase et al., 2023; Kendon et al., 2023), is shedding new light on the interaction between future climate change and regional hydrological responses. Convective-scale regional models can also be deployed for shorter, targeted purposes. For example, by focusing downscaling onto event sets where such high regional resolution is expected to add value to coarser scale models, or by sub-selecting global projections that allow a broad range of climate hazards, needed for robust adaptation, to be simulated regionally at high resolution.

While the combination of high-resolution coupled global climate models (~10-20 km in the atmosphere and ~0.1° in the ocean) and convection-permitting regional climate models (~1-3 km) is computationally demanding, the potential to deliver radically new findings and policy support, at scales required by national and regional planners, means they are an increasingly important input to national climate scenarios and climate services. This is particularly the case with respect to extreme weather events. In the next phase of CMIP and CORDEX, we propose increased collaboration, as well as increased data and knowledge sharing, between high-resolution global climate models, convection-resolving regional models, and statistical/ML-based downscaling, with the goal of producing a coordinated ensemble of state-of-the-art, high-resolution global and regional projections. We further recommend the resulting (global and regional) projections are used to drive a range of impact models (e.g. in ISIMIP, AgMIP and FishMIP). As the future impacts felt by natural and human systems is not only dependent on climate change, but also on the direct human forcing of climate arising from the underpinning scenarios themselves, it will be important to also represent these drivers at high spatial resolution. The resulting set of climate change and impacts data will be of enormous value to national climate change impact assessments, adaptation planning and climate services. To maximize the quality and consistency of this multi-scale, multi-method data set, it is important systems are developed and employed to support careful evaluation of the cascade of information across methods, scales, and regions, as well as from climate to impacts, highlighting both value-added and consistency-lost across the entire chain.

## 4.3 Global Storm Resolving models and the path to global km-scale

Global models with grid spacing in the range 1-10km are often referred to as Global Storm Resolving Models (GSRMs, e.g., Hohenegger et al., 2020; Judt et al., 2020: Caldwell et al., 2021). GSRMs running at ~3-5km global resolution currently achieve a throughput of ~0.5 simulated years per day (SYPD), with an aim to reach 1 SYPD in the coming years. GSRMs originated within the international DYAMOND initiative (Stevens et al., 2021) and the GRSM community are currently designing year-long experiment protocols (Takasuka et al., 2024, submitted). In addition, within the EU-sponsored Destination Earth (DestinE; Wedi et al., 2022) two coupled GCMs have run a reduced HighResMIP experiment (for the period 1990 to 2040) with grid spacing of 5km.

Examples of scientific highlights realised by GSRMs include; a realistic representation of the interannual frequency of Tropical Cyclones (TC) in major basins, comprising a realistic distribution of all severity categories (Judt et al., 2020), as well as realistic representation of the rate of TC intensification, possible as resolutions reach 3km or better. Recent comparative studies among km-scale ocean models show large-scale features that affect the storm tracks and air-sea coupling (e.g., Gulf Stream separation) are more consistent in these models than in coarser resolution ocean models. Internal variability is also substantially larger in eddy-rich models (Chang et al., 2020; Jüling et al., 2021), including stronger SST responses to AMOC variations. In terms of coupled phenomena, realistic representation of the North Atlantic storm track has been shown to be sensitive to resolution of the ocean mesoscale, including instantaneous (eddies) and climatological features (western boundary currents) (Moreno Chamorro et al., 2022). Representation of the full spectrum of precipitation processed by cyclones, including their frontal structures, organised convection, such as Mesoscale Convective Systems and squall lines are generally more realistic as model resolution is increased (Vellinga et al., 2016).

Many of these achievements have been in the realm of convection-permitting Regional Climate Models (see section 4.2) for the past ~5 years. GSRMs offer the additional value of being able to simulate upscale effects from small scales onto larger scales, e.g. how the Hadley and Walker circulations are affected, including meridional transports of energy, as well as implications for global teleconnections, mediated by atmospheric wave propagation. Many of these achievements were realised thanks to the development of new dynamical cores capable of reducing the total number of computations, by use of uniformly spaced global grids, or by models running more efficiently through advanced numerical schemes in time and space, and by exploiting multiple parallelisation paradigms on the latest supercomputers, including those equipped with GPUs. With the advent of even more powerful new classes of GPU, such as the NVIDIA Hopper or AMD MI300 series, completing a selection of typical CMIP6 experimental protocols at ~3km resolution, with a total turnaround of order of one year, will soon be possible.

Data output and analysis constitutes a major challenge at these resolutions: output of order petabytes per day are commonplace, and storing multiple ensemble members for centennial-scale simulations is not feasible. Multiple approaches are being tested to alleviate this problem, such as performing the most data-intensive and multi-variate analyses while the models are running, reduced data precision, or holding data on fast disks for very brief time periods to allow immediate consumption by users. Other approaches include the use of hierarchical data layers, which can be output and handled in parallel, with incremental expense, as exemplified by the HEALPIX standard. An ambitious vision for addressing such data challenges, including co-design, co-production, and global access, is provided in the Earth Virtualisation Engines concept (Stevens et al., 2024).

## 5      Increasing collaboration across approaches to improve global and regional Earth system and climate models.

The accuracy of numerous simulated Earth system and biogeochemical phenomena strongly depends on the quality of simulated physical climate drivers (Doney et al., 1999). Examples of such dependencies include, but are not limited to; (i) vegetation growth/loss, terrestrial carbon uptake, and the simulated water cycle; (ii) wildfires and simulated precipitation, soil moisture and winds; (iii) marine productivity and the dynamics of ocean upwelling, (iv) mass loss from marine ice

sheets and regional ocean circulation; (v) global ocean heat and carbon uptake, and representation of deep water formation, (vi) regional air pollution and modes of atmospheric circulation. Conversely, in the real-world, carbon cycle – climate feedbacks (as well as other Earth system feedbacks) change the fraction of anthropogenic $CO_2$ (and other gases, such as $CH_4$ or $N_2O$) that remain in the atmosphere to cause warming, thereby influencing the magnitude of physical climate feedbacks (e.g. water vapour, lapse-rate, cloud or sea ice feedbacks). Furthermore, while an accurate simulation of the mean climate (in time and space), as well as trends in this measure of climate, are important, an accurate representation of variability (in both time and space) of the underpinning physical climate can often be as important for simulating the Earth system response to a changing climate. Such variability is also a critical driver of the impacts of climate change. Regional climate variability, particularly the width of the distribution of such variability (i.e. the extreme tails of future climate distributions), is generally better represented as resolution is increased, both in global and regional models (Wehner et al., 2014; IPCC, Doblas-Reyes et al., 2021; Ban et al., 2021).

High-resolution coupled global climate models can be viewed as the physical core of the next generation of Earth system models, offering an improved simulation of the driving physical climate, including climate variability and extreme events. Collaboration across the development of high-resolution physical climate models, and Earth system models that emphasize enhanced process-realism, needs to deepen both in CMIP7 (with respect to global models, Dunne et al., 2023) and CORDEX (with respect to regional models). Such collaboration can benefit from, and feed into, ongoing efforts under the WCRP LHA Explaining and Predicting Earth System Change (https://www.wcrp-climate.org/epesc), and offers an unprecedented opportunity to bring advances from both areas together to support development of the next generation of Earth system models. Such a meeting point between these two model development paths offers a unique testbed for assessing technological advances (e.g. hybrid-resolution ESMs, Berthet et al., 2019; AI-based emulation approaches, Son et al., 2024), as well as conceptual challenges in Earth system modelling (e.g. quantifying and optimizing the benefits and trade-offs between resolution, complexity and ensemble size). AI/ML-based approaches also have the potential to improve model parameterizations, while potentially also increasing computational efficiency, enhancing the overall projection capability of these models. This needs to be further explored (Eyring et al., 2024a), with increased sharing of methodologies and findings across ML-based, and more traditional (process-based) approaches to model development (Schneider et al., 2024). Increased collaboration and knowledge sharing across these efforts can lead to a step change in our overall ability to provide robust climate information at scales that meets the needs for mitigation and adaptation decision-making (Eyring et al., 2024b).

A number of initiatives are beginning to develop "Digital Twins of the Earth" (DTEs, Bauer et al., 2021; Hoffman et al., 2023), (e.g. the WCRP Digital Earth LHA, https://www.wcrp-climate.org/digital-earths) targeting an optimal fusion of Earth system modelling and observations, to deliver fit-for-purpose and actionable information to society. These approaches combine forward modelling, data assimilation, and machine learning tools with user models designed to answer specific questions. A number of (global and regional) DTEs are beginning to provide samples of km-scale information, with the majority of DTEs to-date being atmosphere-land only models. For application to future climate change, such models presently require sea surface and sea ice boundary condition data (or atmospheric boundary conditions) derived from coupled ESM projections. As DTEs further develop to include other components of the Earth system (e.g. oceans, cryosphere, carbon cycle etc) it will be important they are carefully evaluated against existing approaches to deliver high-resolution future climate information (either via uninitialized projections or observation-initialised predictions). It will also be important to document the uncertainties in DTE projections/predictions arising from different modelling choices, different external forcings and emission scenarios, as well as from internal variability. This is particularly important with respect to predicted or projected changes in future extreme weather events, which by definition are rare occurrences, with low predictability.

702

Only a few efforts to date are trying to develop two key aspects of digital twins; linking inputs to observations and outputs to human systems. In Europe, Destination Earth (https://destination-earth.eu/) experiments with weather and climate twins, down to resolutions of 2.5 km, and aims to make its experimental design respond to user needs, so models store a minimal amount of data, but are re-run on a regular basis, incorporating the latest data requests in each update. In the US, the Department of Energy has tested combining physical models (e.g. the Energy Exascale Earth System Model, E3SM (Golaz et al., 2022)) with human system models, including Integrated Assessment or Energy Grid models. In addition, ultra-high-resolution global storm-resolving models (GSRMs, Stevens et al., 2019; Lee and Hohenegger, 2024) run at 1-5 km resolution may provide further understanding and insights into biases, complementing CMIP7/CORDEX simulations. Increased sharing across the range of modelling communities will benefit all strands of work, improving our combined ability to model the Earth system and deliver robust and actionable information to policymakers and society.

## 6 Improving model simulations of the observational record and key metrics of climate change

To increase confidence in future projections it is important models accurately reproduce the observed historical record. This requirement encompasses multiple variables and timescales, with long-term trends in global mean surface air temperature (GMSAT), including the forcings and feedbacks controlling these trends, of first order importance. In CMIP6 a number of ESMs exhibited EffCS values (of 5°C or greater) that are higher than the 5-95% range, as assessed by multiple lines of evidence (Sherwood et al., 2020). Some of these models also simulated global warming rates over recent decades (~1980 to 2014) greater than seen in observations (Tokarska et al., 2020), leading to suggestions these "hot models" were unrealistic and should be filtered out from climate impact assessments (Hausfather et al., 2022).

Cloud feedbacks are the largest contributor to uncertainty in EffCS. Perhaps surprisingly, CMIP6 ESMs with high EffCS often evaluate better against observations for present-day clouds than earlier or lower EffCS models (Bock and Lauer, 2024; Kuma et al., 2023), and also accurately reproduce recent trends in cloud-radiation when driven by observed sea surface temperatures (SSTs, e.g. Loeb et al., 2020). These ESMs also represent a number (though not all) cloud feedback processes more accurately than earlier models, particularly those related to mixed phase clouds over the Southern Ocean (Jiang et al., 2023). Nevertheless, studies continue to highlight problems across the majority of CMIP6 models with respect to Southern Ocean clouds (Schuddeboom and McDonald, 2021) and, in particular, low-level tropical marine clouds (Konsta et al., 2022), with observation-based constraints of the latter cloud type suggesting an EffCS closer to 3°C (Myers et al., 2021). It is therefore possible some high EffCS CMIP6 models improved one cloud feedback (e.g. mid-latitude, mixed phase clouds leading to a less negative cloud phase feedback) that exposed other feedback errors (e.g. too positive low-level, tropical marine cloud feedback) that previously compensated each other with respect to the total cloud feedback. Such one-sided improvement can result in an increased positive total cloud feedback and high EffCS. Continued improvement in the representation of cloud processes and feedbacks across all relevant cloud types, including exploitation of new observational data and analysis methods, will be crucial for better constraining EffCS in CMIP7 and improving the simulation of historical climate and rates of global warming.

While a number of high EffCS models in CMIP6 simulated too strong global warming over the period ~1980 to 2014, establishing a direct link between EffCS and historical warming is not straightforward. This is mainly due to the confounding role of aerosols, as well as the important role played by natural variability. In CMIP7 historical forcings are planned to be extended to 2022 (i.e. 8 years longer than in CMIP6). Recent studies suggest anthropogenic effective radiative forcing (ERF) has become more positive, by ~50%, between the decades 2000-2009 and 2010-2019, mainly due to a

reduction in the negative aerosol ERF (Jenkins et al., 2022; Hodnebrog et al., 2024). This change has been accompanied by almost a doubling of the GMSAT warming trend between these two decades. Jenkins et al. (2022) suggest that while some of the increased GMSAT trend is very likely due to reduced aerosol cooling, long-term variability in ENSO may also contribute. Modelling studies by Wang et al. (2023) further suggest that decreasing aerosol emissions may outweigh decreasing $CO_2$ emissions in terms of their impact on warming and climate extremes during the path to global net-zero carbon emissions. Kang et al. (2023a, b) suggest the SST pattern observed in the Pacific between ~1979 and 2013, which induces a negative cloud feedback term (that is not captured in most coupled ESMs), is linked to cooling SST trends in the Southern Ocean over this period (also not captured in coupled ESMs). They suggest that as Southern Ocean SSTs begin to warm, the tropical Pacific SST pattern may decay, resulting in a more positive cloud feedback and potentially an increased rate of global warming. Understanding and simulating the drivers of such SST trends, as well as their interaction with climate feedbacks and global warming, will be crucial to increase confidence in future projections.

Constraining future feedbacks and evaluating model processes controlling these feedbacks is a difficult challenge. Emergent Constraints, which use a multi-model ensemble to identify relationships between observable Earth System variations and projected future changes, are an attractive way to constrain future feedbacks based on observations (Hall et al., 2019; Nijsse et al., 2020) and thereby reduce uncertainty in future projections. To date, assumed emergent relationships are often simple linear regressions. Machine Learning techniques are a promising route for identifying multi-dimensional, non-linear relationships between contemporary observables and the future state of the Earth System (Schlund et al., 2020) and may therefore improve the constraints on future feedbacks and even allow an evaluation of model processes controlling these feedbacks. An improved simulation of the historical past, combined with improved constraints on key feedbacks and the processes controlling these feedbacks, will increase confidence in ESM projections and improve estimates of key climate change metrics such as EffCS, TCR and TCRE with implications for estimates of allowable carbon emissions commensurate with different policy targets.

Both global and Regional ESMs struggle to accurately represent observed regional climate trends, as underlined for Western Europe by recent literature (Ribes et al., 2022; Schumacher et al., 2023; Vautard et al., 2023). This may be partly linked to poor quality lateral and surface boundary conditions (e.g. most recently from CMIP6 ESMs), but may also be a result of missing, or poorly represented, regional forcings and/or feedbacks in the RCMs themselves (Nabat et al., 2014; Boé et al., 2020; Taranu et al., 2022, e.g. the representation of aerosol-climate interactions or the simulation of regional/coastal SST trends). For RCMs, too short evaluation runs, and lack of adequate calibration strategies may also contribute to these problems. Tackling such weaknesses, combined with development of an evaluation system applicable across the scales and downscaling methods involved, will be important for increasing trust in high-resolution, regional projections that are used in numerous national climate scenarios and impact assessments.

## 7       Sampling and quantifying future uncertainty

Multi-model ensemble projections (MME), such as those from CMIP and CORDEX, sample a number of plausible IAM emission and land-use scenarios. The MMEs often include a small number of ensemble members per individual model, each sampling internal variability (as represented by that model). The MME approach, to a limited extent, also addresses structural modelling uncertainty. The degree this aspect of uncertainty is sampled is ultimately constrained by the resolution and process realism of the models involved, and by the degree of commonality of approaches to representing unresolved and uncertain model processes (Merrifield et al., 2023).

### 7.1 High Impact Low Likelihood (HILL) outcomes.

While such MMEs sample a fraction of the uncertainty in future Earth system change, this sampling is far from complete, particularly with respect to the extreme, low-likelihood end of potential Earth system change. Such responses are referred to as HILL (High Impact, Low Likelihood) outcomes (Wood et al., 2023). While HILL outcomes have a low likelihood of happening, there remains a small chance they will occur. One example would be if the Earth's equilibrium climate sensitivity (ECS) turned out to be ~5°C. While this outcome is highly unlikely (IPCC AR6 quotes the *very likely range* (5-95% probability) of ECS as between 2°C and 5°; see Fig. 7.18, in IPCC, 2021, Ch7, Forster et al. 2021), if it did occur the impacts on society would be extremely large.

HILL events may also occur at lower levels of warming (Armstrong-McKay, 2020) and impact numerous parts of the Earth system across a range of regions and timescales. For example, a HILL event may be triggered if a threshold of Antarctic ice loss is exceeded, which may then accelerate and become irreversible, with consequences for sea level rise and coastal communities (Garbe et al., 2020; Taherkhani et al., 2020). Similar, poorly quantified, and poorly understood, risks exist for other potential Tipping Points in the Earth system, such as collapse of the Atlantic Meridional Overturning Circulation (AMOC, Klose et al., 2023), dieback of the Amazon rainforest (Parry et al., 2022), or rapid permafrost thaw (Turetsky et al., 2020). Tipping points also exist in the natural environment and in society and may be triggered at modest levels of warming. Examples include climate driven species loss already occurring at today's levels of global warming (e.g. first species extinction attributed to climate change; IPCC 2023 SPM), mass mortality in coral reef ecosystems (Donner et al., 2017; Hughes et al., 2018; Hughes et al., 2019), shift from kelp- to urchin-dominated coastal communities (Rogers-Bennett and Catton, 2019; McPherson et al., 2021). HILL events, both in the natural Earth system and society are not only sensitive to changes in the mean climate, but also to changes in climate variability. Increased inter-annual variability can have major impacts on society and ecosystems (von Trentini et al., 2020). Systematic shifts, even in sub-seasonal climate can significantly impact society (e.g. changes in the frequency distribution of hot summer days and nights, and human mortality; Schär et al., 2004).

The signal of natural variability (in models expressed as internal variability across a model ensemble) increases in importance, relative to the signal of human forced climate change, as spatial and temporal averaging scales decrease, and projection timescales become shorter (Hawkins and Sutton, 2009). A consequence of this is that larger ensembles are required to reliably detect a forced climate change signal from an extreme realization of natural variability. The shorter duration and/or rarer the event, the larger the ensemble size likely required to be confident a (forced) signal is outside the range of natural variability. This is important information for reliable and cost-effective adaptation to potential future climate risks. Several groups have produced large ensembles covering the historical past and future (Olonscheck et al., 2023; Maher et al., 2021; Deser et al., 2020), using 50 to 100 realizations, often started from different initial conditions taken from the model's pre-industrial simulation. Such large ensembles are ideal for detecting forced regional changes (as simulated by that particular model) from internal (natural) variability (also as simulated by the particular model). Due to the high computational cost involved, to date such large ensembles are generally based on relatively low-resolution models that do not carry the process complexity of full ESMs. This can limit their overall utility. For example, low resolution models struggle to simulate intense weather events, such as tropical cyclones or extreme precipitation. As a result, their utility for investigating changes in extreme weather is limited, although this limitation could be addressed, for specific regions at least, by building ensembles consisting of both Global and Regional models run in tight coordination.

Recently, single model initial condition large ensembles (SMILEs) have been combined to form multi-model ensembles of SMILEs (Lehner et al., 2020), increasing the sampled uncertainty beyond internal variability to also encompass (to some

degree) structural model uncertainty. Techniques have been developed to optimally combine individual SMILEs, with
different ensemble numbers, to produce an unbiased multi-model SMILE that also considers present-day model performance
in its design (Merrifield et al., 2020). New Machine Learning techniques offer the potential for a more efficient and
comprehensive assessment of the future projection uncertainty space and can be used to guide, and in some cases realise, the
creation of large ensembles, including ones targeted onto extreme event risks (Eyring et al., 2024a).

### 831 **7.2    Internal variability, parameter uncertainty and model structural uncertainty.**

An additional approach for investigating modelling uncertainty is the Perturbed Parameter Ensemble (PPE) (Murphy et al.,
2007). In the PPE approach uncertain, often difficult to constrain, model parameters are varied within reasonable limits,
where possible constrained by observations (Booth et al., 2017). The resulting PPE members can be further filtered to retain
only skilful members in terms of present-day climate and/or historical trends (e.g., Sexton et al., 2021; Peatier et al., 2022).
Recent advances in model calibration (e.g., Hourdin et al., 2021, 2023) will be instrumental in better designing future PPEs.
Using the PPE approach, it is sometimes possible to mimic key measures of future projection uncertainty (e.g. the range of
climate feedbacks and ECS in a CMIP MME) using only a single model (Collins et al., 2011). Applying the PPE approach
across multiple global and regional model systems allows probabilistic regional climate projections that sample a significant
fraction of the future projection uncertainty (Evi et al., 2021). Such approaches support an assessment of regional impacts
sampling uncertainty in the future driving global and regional climate, including changes in climate and weather variability.

In addition to physically based models, advanced statistical methods such as emulators (Meinhausen et al., 2011; Leach et
al., 2021) and Machine-Learning (ML) (Watson-Parris, 2021; Eyring et al., 2024a) are increasingly being used to more fully,
and rapidly, investigate uncertainty in future Earth system change. Emulators and ML methods can be trained either on an
individual model or an ensemble of historical and future projections made by ESMs (Beusch et al., 2020; Nath et al., 2022)
or RCMs (Doury et al., 2022, 2024) and used to investigate a large range of future emission and land-use scenarios, or to
focus on specific aspects of projection uncertainty (e.g. high ECS futures). Process understanding and observations can also
be brought into the emulation process, enabling the resulting emulators to mimic the behaviour of the more complex ESMs
(Séférian et al., 2024), while weighting this behaviour towards better performing models (Beusch et al., 2020; Sanderson et
al., 2017). Statistical emulation approaches are also used to assess the sensitivity of ESMs to uncertain model parameters
(expanding the PPE approach), both for parameterization development (Silva et al., 2021; Rasp et al., 2018) and for
developing and selecting ESMs that combine acceptable present-day performance with constraints on their future response
(e.g. constraining ECS to lie within a specified range (Peatier et al., 2022)). Emulators were used extensively alongside
global and regional projections in IPCC AR6 to deliver observation-constrained future projections (Nicholls et al., 2022).
Emulators and ML tools can enhance the provision of climate information (Pfleiderer et al., 2024) and support
interdisciplinary integration, allowing direct coupling to IAM scenarios and thus supporting cross-working group
collaboration in IPCC AR7 and beyond.

### 859 **7.3    Assessing uncertainty across all the steps in providing actionable climate information.**

The new round of international modelling projects presents an opportunity to bring together the range of approaches and
methods used to assess and quantify uncertainty across IAM models and scenarios, global and regional models (considering
internal model variability, parameter uncertainty and structural model differences), and impact models (both in terms of the
climate forcing used and uncertain impact model parameters). This collaboration should also extend to communities
developing, improving and applying emulators and simple climate models (Séférian et al., 2024). Collaboration across
communities and activities will help increase the range of uncertainty space that can be analysed, and lead to a more
systematic and coordinated approach to uncertainty assessment across the full suite of modelling activities delivering
knowledge and data to climate policy and services. We further recommend significant effort be devoted to the
communication of uncertainty and conversely, communication of what is expected to occur in the future, and the level of
certainty/confidence that can be attached to these outcomes, with the target audiences being climate change policymakers,
planners, and practitioners.

Going forwards, a key demand on the international modelling community, with respect to supporting IPCC AR7 and the
UNFCCC Global Stocktake, will be the development and analysis of realizable future pathways that limit global warming to
the targets of the Paris Agreement. These pathways are likely to include an overshoot of the warming targets and therefore
the need for negative $CO_2$ emissions (i.e. active removal of $CO_2$ from the atmosphere). How these negative emissions will be
realized in practice and what magnitude is feasible, remain open questions. A thorough analysis and quantification of the full
cascade of uncertainty associated with such pathways is an important demand on the science community. This analysis needs
to encompass uncertainty in; how the necessary negative $CO_2$ emissions will be realized (i.e. the mitigation actions
themselves), the response of the carbon cycle to decreasing atmospheric $CO_2$, the efficacy of any $CO_2$ removal in reducing
global temperatures, and the regional climate responses that may arise from such cooling pathways.  In addition,
uncertainties in the (expected) reduction in the societal and environmental impacts of Earth system change, as global
warming is reduced, need to be assessed, and the impacts avoided compared to any impacts arising directly from the
mitigation actions themselves. Along the entirety of this chain of events and responses there is deep uncertainty. The science
community needs to analyse, quantify, and communicate this uncertainty as thoroughly and clearly as possible.

Robust climate adaptation requires information on the range of potential future changes (which represent the climate hazard
in risk decision frameworks). While progress has been made in quantifying global and large-scale impacts arising from a
range of climate change drivers, this has only been partially successful with respect to translating these impacts to the scales
needed to develop local to national adaptation plans. CMIP7 offers an opportunity to more fully include and propagate the
wider $CO_2$-emission driven uncertainties through to local-scale climate information (as outlined in Sect. 3.2).  An equally
important dimension is the role natural variability plays in climate change, especially on the timescale of the next 10 to 40
years (that frames many adaptation decisions). On these timescales and at the local scale, natural variability typically
dominates the forced climate change signal, for example for precipitation and temperature. This information is ever more
critical as society adapts to climate change in a mitigating world, where such mitigation aims to limit the climate change
signal. Large initial condition ensembles are a key tool for understanding and quantifying the role natural variability plays.
The expense (computational, data storage) of generating and sharing Lateral Boundary Conditions (LBCs) required to drive
Regional Climate models has limited the availability of LBC data, and hence the potential for regional scale simulations
(such as CORDEX) to sample the role of regional natural variability in the context of the wider climate hazard space, at
impact relevant scales. Commitments for new LBCs are often made before a simulation's credibility can be assessed and
before any understanding of where the realisation of variability plus feedbacks places a particular simulation in the wider
potential projection space. There will be value, therefore, in exploring iterative approaches between ESM and regional
modelling groups to identify optimal ESM simulations to be rerun for LBC generation.

Statistical downscaling may provide the most effective route to link wider ESM projections to what they imply at the local
level (Gutiérrez et al., 2019), as these approaches are not restricted by the limited availability of LBCs. Emerging Neural
Network Machine Learning techniques trained on existing regional (RCM and Convection Permitting RCM (CPM))
simulations, are showings promise in capturing spatial and temporal climate change, at local scales, based on large scale
drivers simulated by ESMs (Baño-Medina et al., 2021; Doury et al., 2022). Whilst there is still work to be done (e.g.
achieving multi-variate coherence (González-Abad et al., 2023), transferability to other ESMs (Baño-Medina et al., 2024),

and building frameworks to verify ML downscaled results), their emergence is likely to transform how the science community provides local scale climate Information, as they allow the production of this information to be determined by realisations that can inform on the range of local scale climate hazard (bottom up) rather than the limited availability of Earth system model LBCs (top down). ML-based downscaling therefore has the potential to translate coarse-scale Earth system model output directly to spatial scales of utility for impact models, impact assessment and local adaptation planning (Eyring et al., 2024b). Such developments can be transformative in other senses, too. For example, given adequate prior ESM to RCM/CPM training data, CMIP7 has the potential to be downscaled almost as soon as the ESM simulations are completed, something which could help inform, for the first time, IPCC AR7 with consistent global and regional projection data, and associated impact simulations (see Sect. 2). Similarly, ML may offer ways to address the prohibitive storage costs of conventional high resolution local data by enabling the availability of such data on demand based on large scale variables (which are much cheaper to store). Ultimately, incorporating Machine Learning into the production of high-resolution regional climate information is likely to open further benefits due to the flexibility such tools enable. For example, ML downscaling will be amenable to approaches that use observations to bias correct the regional data, directly. Similarly, as insights from new modelling (e.g. resolving convective scales, interactive atmosphere-shelf sea-wave models) come online, ML downscaling tools may be able to produce new high resolution regional climate data reflecting these insights, if modelling experiments are designed to inform the required ML training.

## 8        The underpinning technological infrastructure

The ambitious science and science for policy aims discussed in this paper cannot be realized without a state-of-the-art underpinning computational and data infrastructure, supported by experienced personnel. Our recommendations require the co-design of certain experiments, followed by the production, quality-control and sharing of numerous datasets from a diverse range of modelling systems, between producers and a heterogenous set of consumers separated in time and space. An aspiration for IPCC AR7, as described earlier, is to deliver a coordinated and coherent set of data from across the most recent IAM scenarios, global projections (CMIP7) and regional downscaling (CORDEX), as well as impact model results based on these scenarios and climate forcing. To achieve this will require more efficient and rapid sharing of both requirements and data across all communities, including where feasible user communities. We therefore stress the need to improve the underpinning infrastructure ecosystem that supports these modelling efforts to enable the co-development of suitable experiment protocols, followed by the production, evaluation, and exploitation of datasets, which themselves can be used as input to other simulation workflows, with different production, validation, and exploitation cycles. This will need to be realized for far more numerous and larger volume datasets, and across a broader and more disparate set of requirements and communities than was previously the case.

CMIP6, like CMIP5, benefited from a globally coordinated data infrastructure, the Earth System Grid Federation (ESGF), linked to a large array of other important and necessary services (Balaji et al., 2018). The CMIP6 ESGF is now more than a decade old, largely not maintained and is therefore not fit for the scale of the challenge outlined above. The array of services linked to the ESGF include: standards-based data, model and experiment descriptions; citation and errata services for simulation data and derived products; and data quality control procedures (addressing the presence of required data, standards compliance etc, not to be confused with procedures for assessing the scientific quality of the data). The data infrastructure itself needs to support systematic (and efficient) simulation evaluation, and support replication of data from source to "super-nodes" that can host large volumes of multi-model data and provide sufficient local computational resource to allow analysis with minimal requirement for data movement (Eyring et al., 2016). Local computing services will need to include both specific "well known" computational services such as those necessary to generate on-demand statistics, and

those necessary to support user-generated analysis pipelines that may include AI and ML techniques. To realize the ambitions outlined in this paper, the volumes of data that will need to be hosted at such super-nodes will be significantly larger than for CMIP6, and the services will need to be easier to navigate for a more heterogeneous community, extending beyond the modellers and analysts of earlier CMIP cycles.

There are several activities underway that aim to address some of these requirements. Notable amongst these are the development of reusable evaluation and analysis workflows such as ESMValTool (Eyring et al., 2020; Righi et al., 2020) with the goal of fully integrating these into the CMIP publication workflow (Eyring et al., 2016b), the democratisation of the use of cloud computing via Pangeo (Abernathy et al., 2021), the use of new data formats such as HealPix (Chang et al., 2023), and the development of new technologies aimed at a future ESGF (Hoffman et al., 2022). However, there are also significant areas where little or no development is underway. These include enhanced documentation, errata, and citation services, many of which are relying on best efforts and need dedicated investment and effort in new techniques and modes of deployment. Considerable work will be required to bring all of these strands together into a coherent system that can be deployed and supported world-wide and sustained throughout the next IPCC cycle (and beyond).

This new ecosystem will need to support and coordinate efficient methods for data reduction and sharing, cross model analysis and evaluation, with an emphasis on bringing together existing and new observational and reanalysis datasets, models, emulators, and advanced analysis tools for rapid and in-depth analysis and exploitation. The new system will need to interface with other major data holdings, for example those of the WCRP Lighthouse activities[1] (Flato et al., 2023), the Destination Earth[2] data holdings, the existing ISIMIP data repository[3], the Copernicus Climate Change Service (C3S)[4] and new data holdings that may arise from the EVE (Earth Visualization Engines)[5] initiative. It will need to conform to FAIR (*Findable, Accessible, Interoperable, and Reusable*) principles (Wilkinson et al., 2016) and meet the needs and requirements arising not just from CMIP7, but from the range of communities involved in IAMC, CORDEX and VIACS/ISIMIP. Critically, the system will need to be fully supported by dedicated data managers, capable of addressing community questions pertaining to data quality, model and data documentation, as well as supporting users of embedded infrastructure tools to facilitate the rapid use and reuse of data and tools across communities. It is this rapid use and reuse that will deliver the internal consistency, across models and research communities, that is key to the transformative impact expected for international climate policy from the science and modelling efforts proposed in this article.

## 9       Summary and recommendations for the way forward

Over the past three decades, internationally coordinated modelling projects have delivered a wealth of simulations, data, and scientific knowledge to support policy actions addressing climate change mitigation and adaptation. As a new round of these projects start up, and a 7th IPCC assessment cycle begins, we have reviewed how these projects collectively have delivered science support to international climate policy. We propose a number of science, technology and collaboration priorities that we believe these projects should jointly focus on over the coming decade. Progress in these areas will increase the quality and utility of science support to climate policy, while also increasing our understanding of Earth system change, including the impacts on society and the natural world, as well as our ability to model such future changes and the associated impacts.

---

[1] https://www.wcrp-climate.org/lha-overview
[2] https://destination-earth.eu/
[3] https://data.isimip.org/
[4] https://cds.climate.copernicus.eu/
[5] https://eve4climate.org/

One key proposal is for the involved modelling communities, spanning integrated assessment, scenario generation, global and regional Earth system modelling, regional downscaling, and impacts modelling, to work much more closely together during the next round of projects, with an aim to deliver a coordinated set of scenarios, projections and impact assessments all based on the same underpinning socio-economic and mitigation scenarios and using the most up to date model configurations. This will significantly improve the quality and consistency of scientific knowledge available to the upcoming (AR7) and future IPCC assessments, as well as to the 5-yearly UNFCCC Global Stocktakes. Building on interactions developed over the past 5-10 years, and proposals for simulations supporting international climate policy to become more operational in structure, the time is right to actively develop a tighter and more efficient set of links across the relevant modelling projects. Realizing this ambition within the AR7 timeframe is likely not possible. Nevertheless, significant effort to achieve such internal consistency and efficient sharing of data, knowledge, and personnel, will lead to future workflows better suited to fully realizing this ambition. In addition, we highlight the need for impact models to receive more detailed information (disaggregated, spatially and by sector) on the socio-economic assumptions underpinning the IAM scenarios. Conversely, increased effort is required to allow knowledge of projected future climate impacts, and the likely societal responses to these impacts, to be iteratively incorporated into the generation of emission and land-use scenarios. Thanks to CMIP5 and CMIP6 cycles, there is an increasing set of well-established links between IAM scenario production teams, Earth system modelling groups, CORDEX downscaling teams, and impact modellers, with the majority of the modelling in these activities using a common data infrastructure system. These established connections and shared infrastructure make the potential for a more efficient, inter-connected workflow across all these activities a real possibility in the coming years.

The programme of work we outline addresses numerous key knowledge gaps, several of which were highlighted in IPCC AR6 (IPCC, 2021). Given the increasing number of ESMs capable of running in $CO_2$-emission mode, including simulation of the coupled climate and carbon cycle, as well a range of other Earth system phenomena, combined with an increasing number of coupled GCMs running for centennial timescales at ~10km resolution, we believe many of these knowledge gaps can be successfully addressed over the coming decade. Exploitation of CMIP6 was identified as limited in AR6, pointing to a need to support and better focus coordinated international modelling projects, including links between projects. Plausible overshoot scenarios that return to the Paris Climate targets by the end of the century or later (e.g. by 2130), were limited in CMIP6 and need to be a greater focus in CMIP7. To address this, it is crucial ESMs are extended to allow a more thorough assessment of the efficacy of proposed land and marine $CO_2$ removal techniques in reducing atmospheric $CO_2$ and driving global cooling, while accounting for potential Earth system feedbacks (IPCC 2021; Canadell et al., IPCC 2021). ESMs need to be capable of assessing both $CO_2$ and non-$CO_2$ feedbacks during overshoot (e.g. a changing efficiency of $CO_2$ uptake by natural reservoirs as $CO_2$ is removed from the atmosphere, or methane release into the atmosphere from wetlands or permafrost (Canadell et al., IPCC 2021)), as well as the potential for, and consequences of, rapid change in key Earth system components during overshoot, such as ice sheet loss or forest dieback (Canadell et al., IPCC 2021; Fox-Kemper et al., IPCC 2021). In addition, interactions between $CO_2$ warming and trends in aerosol emissions need to be thoroughly assessed, so the impact of decreasing aerosol emissions on the near-term rate of global warming and achievability of the Paris targets can be better quantified. Such analysis needs to be complemented by analysis of the (societal and environmental) impacts of a warming overshoot, the degree of reversibility of these impacts once cooling to a target level is achieved, and the impacts resulting from long-term stabilization at a target warming level (assuming it is warmer than today). The majority of IAM scenarios, designed to realize the Paris Agreement, assume extensive deployment of land-based (and in a very limited number of cases, marine-based) atmospheric $CO_2$ removal technology. The direct impact of these mitigation actions on society and the environment needs to be assessed and contrasted with the impacts avoided from the resulting reduction in global warming. An additional set of approaches to limit the magnitude of future warming, referred to as geoengineering, are increasingly discussed in policy circles and the media. The most widely known being Solar Radiation Management (SRM;

Lawrence et al., 2018; Visioni et al., 2023). While there remain concerns around the safety and governance of such actions, it is increasingly important the research community actively assesses the efficacy of these approaches, including the risks and potential consequences of deployment of this technology at the scales required. Projections beyond 2100 were not comprehensively covered in CMIP6 (Chen et al., IPCC 2021). This is important for understanding committed changes and the consequences of long-term stabilization at temperatures warmer than today. This is particularly acute with respect to sea-level rise (Fox-Kemper et al., IPCC 2021), with Antarctic and Greenland ice sheets representing the largest uncertainty in future sea-level projections. It is vital these systems are better modelled in CMIP7 and beyond.

More accurately simulating the observed, historical evolution of the climate system (i.e. reducing systematic model biases), including the representation of the forcings and feedbacks driving the observed warming, is crucial for increasing confidence in model projections and for maximizing the use observations in model improvement. Associated with this, we advocate the use of new approaches (for example, combining Machine Learning and Emergent Constraint techniques) to enable more extensive use of observations to constrain model projections and future feedbacks. A key requirement remains improved constraints on key metrics of Earth system sensitivity (e.g. EffCS, TCR, TCRE and the Regional to Global Warming ratio) and that models accurately simulate these metrics, including the processes underpinning them.

Due to their exceptional impact, we highlight the need for improved knowledge of, and ability to simulate, extreme weather events, including potential future changes in such events. We further stress the importance of assessing the impact of extreme events on society and the environment, considering the level of uncertainty inherent in projections of such rare events. This requirement also extends to the modes of climate variability that extreme events develop within (including natural variations, future changes and extreme realizations of these modes). Looking towards the next generation of Earth system and climate models, we propose significantly increased collaboration across communities investigating enhanced Earth system process realism, those working on increased model resolution, and improved physical parameterizations, as well as groups working on ML-based hybrid modelling. Increased collaboration across these communities will optimize findings from each approach for development of the next generation of Earth system models. This recommendation holds equally for global and regional models, including collaboration between these communities.

With respect to uncertainty, in future emission scenarios, in Earth system change, and in the impacts, we propose extensive collaboration across the range of approaches addressing these issues. Wherever possible work should assess, quantify, and emulate uncertainty as it propagates through the stages of IAM scenarios, ESM projections, regional downscaling, and impact simulations, so a more complete assessment of total uncertainty can be provided to policymakers. An additional consideration is to better quantify what can be predicted (based on model predictions started from observed initial conditions) versus projected (changes in future climate statistics relative to past or present statistics resulting from external forcing). An important challenge in this area is to accurately quantify the level of predictability at different time and spatial scales, for different variables and regions. We highlight the need for improved modelling and assessment of potential High Impact Low Likelihood (HILL) outcomes, with the possible exceedance of tipping points in the Earth system, in the environment, or in society, being of critical importance. Given there will always be some level of uncertainty in the future climate, it is important to focus on the communication of this uncertainty, or possibly more importantly, communication of what is expected in the future and with what level of confidence. This is a key area in the science-policy interface.

The transformative goals outlined in this paper require the support of a robust, efficient, and internationally connected infrastructure. While components of such an infrastructure exist, much work is needed to design, build, deliver and sustain an integrated system that meets the objectives outlined here, and maximises the benefits of existing initiatives and

investments. The resulting infrastructure must exploit common tools and standards and be designed and delivered with both a long-term perspective and a well-trained workforce. It will need to handle increasing volumes of data, support the use of new techniques for data analysis (such as remote analysis of big data using ML and AI techniques), and facilitate the easy exchange of data, knowledge, and analysis tools. Without such an infrastructure, many of the aims outlined in this paper will not be met in a timely manner, if at all.

Finally, to expand the reach and benefits of international modelling, including the uptake and use of model simulations, to a more global scale and thus deliver underpinning scientific support for global climate policy, there is an urgent need for increased involvement of Global South scientists. WCRP leads a number of important efforts in this area. These need to be ramped up significantly and put on a sound long-term footing. Given the global nature of the climate crisis, that the impacts are, and will continue to be, most strongly felt by Global South countries, a globally inclusive response is a necessity. This makes both scientific sense (to draw on local expertise for understanding and predicting local Earth system change and its impacts), as well as political sense (climate policy is generally better tailored to a specific country's needs if it is based on local expert advice that is accessible over the long-term). We (a group of European scientists) encourage our governments and funding agencies to provide sufficient, long-term support to further develop and maintain a strong and globally inclusive scientific collaboration over the coming decades.

**Author contribution**

All co-authors provided ideas and comments to the manuscript. CJ, HJ, SJ, BNL, RS, TK, KF, BS, BB, SS, DVV, HH, EOR, FA, MR, PF, PLV, VE and PC conceived and developed the original ideas and recommendations in the paper. CJ and HJ wrote the paper, with regular input from the 17 other people listed in the first 19 co-authors and periodic input from all other co-authors.

**Competing interests**

Two co-authors are on the ESD editorial board: Roland Seferian and Richard Betts.

**Acknowledgements**

CJ, RS, BS acknowledge funding from the European Union Horizon 2020 project ESM2025 (grant number 101003536). BN, SJ, FA, VE acknowledge funding from the EU Horizon 2020 project IS-ENES3 H2020 project (grant number 824084). TK, CJ, KF and HJ acknowledge funding from the Horizon Europe project OptimESM (grant number 101081193). PC, VE and PF acknowledge funding from the European Union Horizon 2020 project 4C (grant number 821003). MJR, PLV DVV and HH acknowledge funding from the European Union Horizon Europe project EERIE (grant number 101081383), SS acknowledges funding from the Horizon Europe project IMPETUS4CHANGE (grant number 101081555), VE acknowledges funding from the Horizon 2020 European Research Council (ERC) Synergy Grant USMILE (Grant Agreement No. 855187). BBBB and HH were supported by the UK Met Office Hadley Centre Climate Programme funded by BEIS and Defra. NSF is funded by National Institute for Health and Care Research award NIHR204850 and Natural Environment Research Council grant number NE/Y503319/1. Discussions that led up to and supported the development of this paper occurred in all of the above projects, as well as in the COST Action CA19139 PROCLIAS (PROcess-based models for CLimate Impact Attribution across Sectors), supported by COST (European Cooperation in Science and Technology; https://www.cost.eu).

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
