# Peer review of "Bringing it all together: Science priorities for improved understanding of"

_EGUsphere, 2024_

## Community Comment (CC1)

**Comments on:** *Bringing it all together: Science and modelling priorities to support international climate policy.*

This is a timely article, and I welcome its use of the *EGUSphere* to facilitate an open and accessible discussion on important topics and the invitation to comment. If others engage with reasoned opinions, and if the authors respond to these in similarly reasoned ways, the manuscript can make a valuable contribution to the development of scientific priorities.

As it stands, the article is an imperfect assemblage of what appears to be the individual research programmes of its European authors who collectively have considerable experience in a particular sphere of climate modelling, and its relationship to policy   This is a useful starting point.  However, in many places the paper reads as if it is: (a) a scientific assessment; and/r (b) somehow speaks for a group larger than the authors themselves.   These deficiencies are compounded by an insufficiently critical analysis both in regard to the authors' specific past experience, and of important initiatives by large-segments of the community that precede the submission of this manuscript.

On who the article speaks for: The word international is used nearly three dozen times, mostly in the sense of talking about the needs or experiences of either an "international modelling community" or an "international policy community".  This results in the impression that the authors are trying to speak for these 'communities'.  Even if such communities existed, how can the author's purport to represent them? Likewise, the institutional affiliation combined with the articles normative tone can give the impression that the authors are speaking for the institutions they are affiliated with.  As someone who can speak for my institution, I can say that those authors affiliated with the MPI-M are not representing an institutional view, as we were never consulted on these issues.

A nod is given to the Global South (note both words should be capitalized), quite literally as an after thought in the last paragraph.  Given the exclusively western European author list many will see this as tokenism.  The authors cannot change who they are, but the deficiencies as might occur in the presentation of the research programme from such a limited set of experiences should be more explicitly acknowledged and ways for addressing them suggested.  EVE, which was built from a much more diverse (but still far from perfect) group of scientists, technologists, and practitioners, presents ways to address these issues, by developing state of the art infrastructure in under-resourced regions, this and other proposals could be a basis for a specific discussion of the issues.

Simply by clearly framing the article as a commentary developing the ideas and views of its conceptualizing European authors, and acknowledged their shared history (which I presume, in leading and working together on European projects funded with the objective of  supporting successive past phases of CMIP), would go a long way to addressing the above issues.

The above brings me, to another critical point.  The article emphasizes the importance of coordination, as if this can happen by force of will.   Most of the ideas, and challenges it puts forth are real, but could very easily have been written ten, or even, twenty years ago.  The idea that we need more coordination with scenarios, more rapid throughput to users, more ensembles, higher resolution, better models, better uncertainty quantification, etc are all good and fine, but not new.  Decades of national and European projects, encompassing hundreds of millions of Euro have been conducted under the pretense that they would overcome these challenges.  CMIP6 was estimated by the CMIP panel to have cost €3 billion of funding, yet even its most eloquent advocates have struggled to communicate the scientific accomplishments that merited the effort (Lamarque 2022, Meehl 20023). Given this, the question become: What went wrong and why can it now go right?  A starting point to answering this question is offered be the many recent published analyses. (see e.g., Stevens (2024), Palmer and Stevens (2024), Jakob et al. (2023), Slingo et al., 2022).  Their answer as to what went right and what has changed are as follows:

- Copernicus Climate Change Services
- Destination Earth
- EVE
- Computational capacity enabling qualitatively new approaches to modelling.
- Machine Learning (but not in the sense it is referred to in this article).

Among these, DestinE and resolution are the most substantial, and EVE the most ambitious.

The authors might object that each of the above is mentioned in their article, but the first three only superficially so.  There is expertise on the author team to treat these proposals and projects more seriously.

Resolution is substantially addressed, but the conclusions drawn are in dissonance with the evidence presented.  Likewise, ML is mentioned many times (more than 20) but still as an undeveloped promise with few concrete results, and even in this realm the cutting edge applications as discussed in a series of review article (e.g., Hoefler et al., Bauer et al., Bauer et al) are overlooked.

I would prefer for the authors to build on their vast experience, in coordinating and contributing to projects like ESM2025, CRESCENDO, CMIP, to critically review what was achieved, what went wrong, and what is different now? Doing this in the context of ongoing activities (DestinE, NextGEMS, EERIE) as mentioned above would strengthen the article and give the readers, and those who might draw different conclusions, a line of reasoning that can be discussed, and hopefully resolved.

A lack of specificity poses another problem, in that the article is unclear as to whether it is arguing to continue the practice of providing a climate service (advice to the international policy community) using a research infrastructure, or if it is arguing for an operational activity as has been proposed in a number of high-level commentaries.  This is a fundamental point as the literature claims that the failures (challenges) that the article articulates are a direct consequence of trying to provide an operational service using a research infrastructure (Bauer et al., 2021, Slingo et al., 2022, Jakob et al., 2023 and Stevens et al., 2024 Stevens 2024).  Do the authors think this analysis is incorrect?  Are they really of the mind that the challenges their manuscript poses can be met by coordinating a research infrastructure?  If so it seems reasonable to expect them to explain why this hasn't worked over the past two decades, and three CMIP cycles, and what has now changed that will make it more effective in a fourth attempt.  Alternatively if they indeed are proposing an operational service, then more explicitly building upon and extending the ideas presented by Bauer et al., Slingo et al., 2022, Jakob et al., and Stevens et al., 2024, would make this more tangible.  For an article that advocates working together, the absence of a critical discussion of open proposals and ongoing activities by their colleagues makes the authors seem uninformed.

**The modelling challenge:**

 In addition to addressing the challenge of processing information, from scenarios to global models, to impacts, and back, the article again emphasizes the need to improve the suite of tools, particularly the modelling, that is central to this information processing.  I suspect there is no disagreement on this point. The real question is however, what strategies are likely to be most successful, and why now?

Throughout our careers we have undoubtedly all heard that models need to improve.  I have reviewed and participated in scores of projects that are motivated by the need for improved parameterizations.  A common approach has been to learn from high (km-scale and finer) resolution.  I've organized and led some of the largest modelling exercises and field campaigns dedicated to these activities.  As touched on by my recent CMIP essay (Stevens 2024), the unfortunate reality is that this has not produced any real examples of improvements to parameterizations that systematically improve all models, i.e., the same change across a score of models that lead them all to be better.[1]   The efforts for sure have greatly deepened our understanding of how low clouds work. Likewise we now appreciate the important ingredients for representing rainfall, and extremes, as well as for a host of other processes.  In most cases we understand the failure of fundamental assumptions common to efforts to apply parametric solutions (e.g., lack of scale gap, assumption of equilibrium) so as to preserve the computational economy of coarsely resolved approaches.  These findings end up reinforcing the null hypothesis, which is that parameterization is a model specific form of error compensation, so that particular instances of model improvement are likely to be an example of overfitting (how else to explain the Zhu et al (2023) results.  This means that even the meager progress across CMIP phases is, where not related to resolution, not robust  It means that overfitting is real, which makes me (and I suspect many others) skeptical that machine learning will be more helpful than misleading as a way to better (rather than just more efficiently) represent the effects of sub grid-scale processes.

Hence the question becomes what has changed?  The main thing is the ability of increased computational capacity to tap scales that allow for structurally different simulation systems. This article has, buried within it, many insights that reinforce this point, through its highlighting of a small subset of the numerous and specific examples of how storm-and-eddy resolving capabilities improves simulation quality. These examples stand in contrast to unspecific references to the importance of parameterization development, machine learning, or emergent constraints. And while deep in its bowels the article recognizes the importance of resolution for representing impacts (downscaling) and physics (upscaling), it still misses the
* * *
[1] Across all parameterizations, and apart from the long-tail mixing introduced by Louis in the late 1970s,  or perhaps the Gent-McWilliams eddy parameterization from the 90s, it is hard to identify a single parameterization development that led to systematic improvement across models. Even the systematic benefits of GM are disputed.

key role resolution plays in linking to data (observations). More importantly, its message is systematically undermined by repeated references to the need to improve resolution only to the limit that parameterized models can comfortably access, i.e., (10-20 km) scales rather than the breakthrough scales of 5-10 km and finer.

What is the rationale for this? Are too many of the authors unaware of the state of the art? DestinE has already begun to broach scales where we can represent new physics. Multi-decadal to centennial multi-model simulations can be run with full earth system models (carbon cycle, dynamic vegetation, aerosol are already coupled and ice-sheets pose no particular difficulty) at scales ranging from 2.5-5 km globally. The first scenarios are running on Lumi now, at 5 km, globally, with a throughput of 0.33 SYPD and larger, on about 5% of that machine. In the coming year machines twice as large will become available in Europe. This means that dedicating just one of the existing Tier-0 machines would deliver more than 2000 simulated years per year. Wait a year and we will have twice as many resources. But even now the computational capability is enough for small ensembles of multi-model multi-decadal projections (30-50yr) at 5km (horizontal) grid spacings. The computing capacity is there, as are the models, albeit not all models. Do the authors really believe that the problem is not important enough to use these technologies, or that before using them we need to wait until everyone's model can use them?

This doesn't mean that km-scale global models are the only thing that needs attention, but by being more specific as to what this capability means for existing approaches, and how these approaches could be adapted in light of the new capabilities, any claim to support existing approaches sounds self serving. From my point of view regional models will be very important for wide-scaling (i.e., exploring scenarios to enable machine learning interactivity), and models with the much coarser (20km-200km) resolution will continue to be important for: multi-centennial to millennium scale simulations, for interpreting the results from more physical (highly-resolved) models, and perhaps to improve global scale ensemble inflation techniques with machine learning. Unfortunately the article leaves the reader, at least this reader, with the impression that we should rather double down on a hand whose best cards were played long ago, i.e., in CMIP3.

**More minor points:**

1. the article gives the impression that CMIP is synonymous with Scenario MIP. And sometimes by only mentioning specific MIPs when not referring to the DECK or the scenarios it reinforces the idea that CMIP is about the scenarios, and the 'community' MIPS are an afterthought. Although this thinking is strongly reflected in the way CMIP presently presents itself, this assumption should be made explicit, or avoided.

2. ensembles are often mentioned, but it is not specifically explained why new ones are still needed. Now that we have a reasonable characterization of ensemble variability, what is to be learned by running new ensembles. Does ensemble variability across large-ensembles differ greatly? If so do we understand why, and how new ensembles might help us better identify the correct answer?

3. considerable attention is devoted to uncertainty, but the words ring empty when there is no strategy to estimate it, and there is no effort to separate it from model spread. Confusing uncertainty with model spread and the spread of model errors does the reader, and our science, a disservice.

4. the focus on negative emission scenarios is interesting, but I missed the question as to why we specifically need to run ensembles of ESMs to address this issue. Before embarking on this endeavor one should answer the question of what the emulators and well tuned simple box models miss and, for this process, what an ESM adds, and why. And, for what it adds (including for the trivial answer of unreliable spatial patterns) why should we believe it. This raises the broader question as to why aren't emulators enough for policy?

5. not addressing the really pioneering advances of the Japanese colleagues in understanding the role of resolution for important atmospheric processes is an enormous oversight. A starting point is the Satoh et al., review article, the contributions it cites, and some since them (e.g., Takasuka et al.), merit more attention.

In retrospect, the minor points (particularly points 2, 3 and 4) constitute a further major point, which is that the article risks being seen as championing a culture of repetition, for repetition's sake. If we do things again, there have to be specific and compelling reasons for doing so, to do otherwise is not scientific.

**Summary:**

The article is timely, taps considerable expertise, but is insufficiently critical and reflective.   My specific concerns can be summarized as follows:

1. Address who the authors are, and how the article should be read, i.e., as a common position statement of the undersigned authors.  Or are they really representing something broader, and if so what and how?

2. Critically reflect on what, based on the authors' experience, went wrong (or right) in the past, what is now different, and why and how does that lead us to expect we can do better?

3. Draw the consequences from the experience that only proven path to better models is to better resolve the underlying physics.  Explain why, given that decades of efforts to address long-standing systematic biases through improved parameterizations have failed — even for cases where we know the answer (e.g., Fiedler et al., 2019) — should they suddenly succeed now?

4. Constructively engage recent proposals (e.g., EVE, Slingo et al, the Scientific Advisory Board of the WMO, the Royal Society Briefs for COP, Jakob et al), which make the case for operationalizing the process of climate information provision for policy and applications.

5. Explain how the scientific community can best benefit and strengthen considerable investments in initiatives such as DestinE and C3S.  (In this regard, and on a specific but important point, the Jakob et al. article is cited, but incorrectly.  Jakob et al. argue quite clearly for an operational activity, not a quasi-operational repurposing of a research infrastructure.)

Addressing these five points, and framing the presentation in a way that is consonant with this analysis would be more scientifically exemplary.  Even as a commentary or perspective the article should strive toward this standard.

Bjorn Stevens,
Max Planck Institute for Meteorology

**Selected (own) References:**

(Apologies for almost exclusively sharing only own references, but an effort to be more comprehensive is, as I see it, the job of the authors.)

Bauer, P., Stevens, B. & Hazeleger, W. A digital twin of Earth for the green transition. *Nat. Clim. Chang.* **11**, 80–83 (2021).

Bauer, Hoefler, Stevens, Hazeleger, Digital twins of Earth and the computing challenge of human interaction, Nature Computer Sci., (2024) in press

Bauer, P. *et al.* Deep learning and a changing economy in weather and climate prediction. *Nat Rev Earth Environ* **4**, 507–509 (2023).

Fiedler, S. et al. Simulated Tropical Precipitation Assessed across Three Major Phases of the Coupled Model Intercomparison Project (CMIP). Monthly Weather Review 148, 3653–3680 (2020).

Hoefler, T. *et al.* Earth Virtualization Engines: A Technical Perspective. *Comput. Sci. Eng.* **25**, 50–59 (2023).

Jakob, C., Gettelman, A. & Pitman, A. The need to operationalize climate modelling. Nat. Clim. Chang. 13, 1158–1160 (2023).

Lamarque, J.-F. Planning for the next phase(s) of CMIP. (2022). Report to Joint Scientific Committee (JSC) of WCRP.

Meehl, G. A. The Role of the IPCC in Climate Science. in Oxford Research Encyclopedia of Climate Science (Oxford University Press, 2023). doi:10.1093/acrefore/9780190228620.013.933.

Palmer, T. & Stevens, B. The scientific challenge of understanding and estimating climate change. Proceedings of the National Academy of Sciences 116, 24390–24395 (2019).

Slingo, J. et al. Ambitious partnership needed for reliable climate prediction. Nat. Clim. Chang. 12, 499–503 (2022).

Stevens, B. A Perspective on the Future of CMIP. *AGU Advances* **5**, e2023AV001086 (2024).

Stevens, B., et al.,: Earth Virtualization Engines (EVE), Earth Syst. Sci. Data, in press, 2024.

Satoh, M. et al. Global Cloud-Resolving Models. Curr Clim Change Rep 5, 172–184 (2019).

Takasuka, D. et al. How Can We Improve the Seamless Representation of Climatological Statistics and Weather Toward Reliable Global K-Scale Climate Simulations? Journal of Advances in Modeling Earth Systems 16, e2023MS003701 (2024).

---

## Community Comment (CC2)

**Bringing it all together: Science and modelling priorities to support international climate policy.**

I welcome open discussions and the opportunity to participate in scientific discussion connected to discussion papers such as these. Here I will provide comments and suggestions, some of which are deliberately critical to motivate refelcations and further discussions - such usually enhance our understanding and improve scientific papers.

My first reaction is that the title of this paper is a bit inappropriate, bringing it all together, as the paper exclusively focuses on modelling work. There is much more than models, although models represent an important aspect of science. Hence, there is not just modelling, but also observations, physics/chemistry/biology/geography and analysis - with a connection to the models.

**L120**. The paper could also ask specifically what kind of information has been used for decision-making and how it has been used. What information should have been used, but hasn't? (Atmospheric $CO_2$-concentrations keep rising despite IPCC reports and COPs; development within climate change adaptation is still slow) Is there really a need for even larger volumes of data, and does the information distilled thereof really improve? (https://DOI.org/10.1038/NCLIMATE3393)

What is meant by "internally consistent" in this case - and why not just consistent? (as used later on)

There is an unstated dilemma - HPC demands a lot of energy and contributes to our carbon footprint. This affects both ESMs, RCMs and AI/ML. Statistical methods, use of mathematics and clever analysis, on the other hand, can alleviate some of the computational needs. Don't forget that.

**L212**: "(RCMs) generate high-resolution regional simulations consistent with the ESM boundary condition…" - this is not true because the RCMs provide more detailed representation of topography and processes which result in different precipitation and hence fluxes than the driving GCM/ESM. GCMs/ESMs and RCMs are often inconsistent with each other (e.g. RCMs tend to generate a different cloud and precipitation climate than the GCM/ESM, and hence give different OLR aggregated over the same region, https://doi.org/10.1175/JAMC-D-20-0013.1). One question is whether this makes a big difference (can perhaps explain some biases). Furthermore, bias adjustment is often necessary, which often renders the results internally/physically inconsistent.

"CMIP simulations are extensively used to inform policy addressing global climate change risks …" - this is perhaps true to some extent, but they have nevertheless not resulted in decisions that meet the Paris accord. This is a big problem. The current global warming does not happen naturally, but is a consequence of human activity. The $CO_2$-concentrations seem to have accelerated after each COP and IPCC report (e.g. the Keeling curve).

For regional climate change, as well as impacts and climate change adaptation, it is important to stress the results from https://doi.org/doi:10.1038/nclimate1562: that decadal variability is both chaotic (non-deterministic) and pronounced on regional scales.

Use the phrase "*carbon capture and storage*" rather than "*negative emissions*" (throughout the paper) - the latter term is meaningless for most people and our usage of terms has a tendency to spill over to the public discourse. The term is also physically meaningless in a literal sense. Perhaps part of the reason why we struggle reaching our objectives with curbing our emissions is that most people don't understand what we say? Also, "*key warming targets*" should be "*key warming limits*" because we don't aim at global warming reaching 2.0/1.5°C, but want to stay below.

**Section 3.2**: Downscaling - use both RCMs and ESD as they are based on *entirely different assumptions* with different strengths and weaknesses, *independent* of each other. Often downscaling is presented involving only RCMs which results in misapprehension of what downscaling entails. There is also a minimum skillful scale associated with numerical models which is the main reason for why downscaling is needed in the first place (also relevant for **L588-595**). Artificial Intelligence, machine learning and knowledge gaps connected to downscaling should also be considered here. E.g. statistical downscaling can be done in many different ways, and different approaches are suitable for different situations.

**L415-430**: Perhaps mention that we can use model results to explore explanations for biases and improve our understanding of meteorological phenomena such as the jet stream, cyclones, atmospheric rivers, and their role in the hydrological cycle? Through a set of numerical experiments and experimental design. One interesting topic could be to explore so-called tipping points (e.g. thawing permafrost og THC) and how different model set-up affects the results.

**Section 4.2**: High-resolution modelling can be used to study phenomena and calibrate statistical models (e.g. testing the fractal dimension of precipitation; Doblet et al. in progress). Climate risks can be quantified by downscaling the shape of pdfs describing a climate variable (e.g. temperature, precipitation, wind speed) in a direct manner, rather than downscaling individual outcomes and then fitting a pdf to the resulting data points. It's probably more robust to estimate risks through downscaling such statistical properties than through downscaling individual outcomes (statistical properties are often very predictable). What we typically seek is the dependency of the shape of pdfs to large-scale conditions.

**L513:** "Machine Learning (ML) offers the potential to reduce long-standing systematic errors in ESMs and enhance the overall projection capability of these models." - such a quick-fix is not reassuring and may hide serious shortcomings connected to the models. There is a lot of hype on AI/ML, but ML/AI has been used in downsclaing since 1999 (e.g. DOI:10.1175/1520-0442(1999)012<2474:TAMAAS>2.0.CO;2).

**L531:** One drawback with ultra-high-resolution global storm-resolving models is that we need large ensembles to sample regional decadal non-deterministic chaotic variations (https://doi.org/doi:10.1038/nclimate1562). We can look at seasonal forecasting which typically involves a large ensemble and not one very high-resolution simulation. There are some unresolved issues regarding seasonal forecasting, such as the optimal set-up of the model themselves (e.g. https://doi.org/10.5194/esd-7-851-2016), which also could be mentioned here as well as decadal forecasting.

**L598**: The phrase "each sampling internal variability (as represented by that *model*)." should be "each sampling internal variability (as represented by that *simulation*)." One specific model can provide many different realisations (https://doi.org/doi:10.1038/nclimate1562). It is important to evaluate the models in terms of their representation of the covariance structure (e.g. https://doi.org/10.5194/gmd-16-2899-2023) and the MMEs in terms of their ability to produce statistical properties that are comparable to the observations (e.g. ERA5).

**L660:** "Such approaches support assessment of potential regional impacts sampling uncertainty in the future driving regional climate, including changes in climate variability and extreme weather." - exactly how do they support and what about model biases and the model's minimum skillful scales? Are there examples of this? References?

**L695:** The phrase 'cascade of uncertainty' is commonly used in the research community, but also a bit misleading. If the uncertainties only cascade, then we should stop at the first step. Of course, the uncertainties don't only cascade, but we also add constraints/information though each step. Hence, it's a question of whether we add more information or more uncertainty (https://DOI.org/10.1038/NCLIMATE3393).

**L702:** The climate sceptics *communicate* uncertainty (doubt), but we *explain* uncertainty. In this case, the main message is nevertheless focussed on certainties, knowledge, as well as their limitations and caveats. If I tell a decision-maker that I have a lot of uncertainty, what should we expect that person to do with that?

**L723**: Gutiérrez et al. (2019) compared a *large number of downscaling methods*, *not wider ESM projections*. However, there are some papers which describe how ESD has been applied to large MMEs (e.g. https://doi.org/10.1175/2010JCLI3687.1, https://doi.org/10.1016/j.cliser.2017.06.013, https://doi.org/10.1175/JAMC-D-18-0179.1).

**L725-rest of the paragraph**: Not just ML, but also statistical downscaling may provide similar results, e.g. through 'hybrid downscaling' (e.g. https://doi.org/10.1175/JAMC-D-20-0013.1).

**L768**: It may be useful to reconsider traditional ways of archiving data, e.g. https://doi.org/10.1016/j.cliser.2017.06.013. At least as an addition to traditional netCDF/CF repositories.

**L782**: It's important to keep in mind the dangers of establishing echo-chambres within a research community. Their existence may be visible through clustering of cited work and omission of relevant references (or examine references herein which don't include the authors of the manuscript). It's hard to keep up-to-date with all the progress and literature, and too easy to walk the same circles over and over again. The paper proposes extensive collaboration across approaches, and there is indeed room for improvement as my comments hopefully will allude to. So, is there a point in thinking differently? (scientists can be quite conservative) For climate change adaptation and decision-making, perhaps put more weight on stress testing or sensitivity testing? Many reports on weather-related calamities involve drought and floods, so why not more discussion on the hydrosphere? (the word 'hydrology' is not even mentioned in the manuscript)

The recent report https://doi.org/10.2777/34601 from the EU may be relevant for this discussion paper.

---

## Community Comment (CC3)

**Comments on Jones *et al.*, 2024**

We welcome the opportunity to discuss Jones *et al.*'s piece "Bringing it all together: Science and modelling priorities to support international climate policy" and submit our comments for consideration by the authors. Our comments are influenced by our perspective as scientists who work at the interface of policy and science. We call attention to the following open questions because greater clarity is needed on these topics to inform current policy discussions. Below, we provide our suggestions for how the modeling community could address these open questions in finer detail and clarity.

Given that we are on the cusp of exceeding 1.5°C, we believe additional thought and investigation are warranted to better characterize the emissions pathways, risks and impacts (reversible and irreversible) that are associated with temporary overshoot. The modeling priorities outlined in your article are an opportunity to provide clarity on (1) what pathways and actions will produce different overshoot scenarios and (2) what the risks and impacts are from those different overshoot scenarios as a function of rate of warming and magnitude and duration of overshoot (see e.g., Reisinger and Geden, 2023).

**1.  Overshoot & Risk Frameworks**

We appreciate how the authors discuss Earth system tipping points specifically in the event of overshoot (Section 3.1 lines 295-305), but propose that the authors explore overshoot impacts more deeply. As a guide, we point the authors to a framework outlined by Reisinger and Geden (2023). In their article, different aspects of overshoot, including peak temperature, overshoot duration, and their integrated sum, are considered for their impact. They ask how these different overshoot aspects are associated with increasing both reversible and irreversible risks. While an overshoot of peak temperature may be temporary, the impacts are not necessarily reversible. We propose that modeling has a role to play in better characterizing these risks and their uncertainties. What modeling approaches and data (observations) are needed to advance this goal? The non-$CO_2$ feedback during overshoot (e.g. methane release from wetlands or permafrost, as noted in line 829) seems to be a major concern but the current ESM capacity in simulating $CH_4$ cycle is low, although some progress is being made with emissions-driven simulations (see e.g., Nzotungicimpaye *et al.*, 2023).

We further propose that both the timeframe of overshoot and the rate of warming are important considerations in this assessment because they may relate to the timeframes over which certain tipping points and HILL events unfold (Ritchie *et al.,* 2023; Lohmann & Ditlevson 2021). The authors already provide important recommendations that HILL events (Section 7.1 lines 604–626) and rapid changes be accounted for in ESMs (Section 9 lines 823-834), and we re-emphasize that these goals be considered within the timeframe of overshoot. To more fully address the role of timeframe, we also suggest the authors investigate both near-term and long-term time windows of overshooting tipping points, such as the next 20, 50, and 100 years,

given that this can inform human adaptation. What emission reductions magnitudes and rates plausibly alter the trajectory of these pathways?

**2. Overshoot & CDR**

Carbon Dioxide Removal (CDR) is a necessary intervention for a temperature exceedance to be temporary, i.e. to be an overshoot. Without CDR or other negative emission or climate intervention, the temperature curve will peak but the accumulated stock of $CO_2$ and ongoing emissions will prevent the curve from bending down. Despite the critical role that CDR plays in our climate goals, its implementation is still largely theoretical. We appreciate the authors surfacing the issue of CDR in their piece (Section 7.3 lines 692-698; Section 9 lines 823-834), and strongly encourage specific focus around its relationship to and constraints on temporary overshoot.

We suggest that the authors consider the recent commentary by Grubert and Talati (2024), where they outline some of the constraints on CDR feasibility, pointing out that the resources and inputs needed for CDR are depletable, and consequently, their implementation will face limits (e.g., finite below-ground storage). They also distinguish between compensatory and actual net-negative CDR, which acknowledges that the amount of CDR available will also be economically limited. We further encourage the authors to consider limitations on BECCS and the time-dependence of emissions and avoiding pathways that result in emissions of irrecoverable carbon (Goldstein et al., 2020).

Given the importance of effective CDR for limiting temporary overshoot, we encourage the authors to explicitly incorporate these knowable limits and constraints into the CDR scenario-making using IAMs (Ramanathan *et al.*, 2021). What do these constraints mean for limitations on the amounts and rates of CDR? How do these feasibility and efficacy limitations place a constraint on the magnitude and timing of overshoot? What are the potential implications for other forms of negative emissions or climate interventions in the context of temporary overshoot and meeting climate goals?

Even assuming future CDR feasibility, there is significant uncertainty over the timing and magnitude of its temperature impact. $CO_2$ removed will have a different temperature impact than if an equivalent amount of $CO_2$ were never emitted (Zickfeld *et al.,* 2023). As a result, preventing $CO_2$ emissions could limit peak warming better than removing an equivalent amount of $CO_2$ through CDR methods. This asymmetry in impact may be due to a variety of factors, including the timing of emissions relative to removals, the effects of co-emitted non-$CO_2$ pollutants, inertia in the climate response, differences in the climate background state, or biogeophysical effects of CDR. Following suggestions made by Zickfeld *et al.* (2023), we encourage the authors to use ESMs to integrate these different factors and increase certainty about the warming impacts of different CDR scenarios (e.g., reforestation vs. DAC).

Distinguishing these non-interchangeable impacts will be essential for clarifying the overshoot peak, timing, and duration of a given pathway.

**3. Clearer differentiation between Emulators and ESMs**

Given our above recommendations for assessing overshoot warming levels and their impacts, we raise for careful consideration the limitations of climate emulators in these assessments. To what extent can emulators assess the risks associated with overshoot and the feasibility and efficacy of negative GHG emissions? Are there fundamental processes missing from emulators (such as HILL) that would limit their value in such assessment?

**4. Clearer differentiation between C1 and C2 overshoot scenarios**

IPCC AR6 WGIII established an implicit near-term temperature goal by differentiating between overshoot scenarios: one category of scenarios that stay below 1.5C in 2100 with no or limited overshoot (C1) and the other category of scenarios with high overshoot (C2). We encourage the authors to develop an ensemble of pathways that would elucidate the potential value of an explicit near-term climate goal.

Additionally, more clarity is needed to differentiate among low and high overshoot scenarios. How can improvements in modeling approaches and scenario design better inform this differentiation and its implications for climate policy? Does the realization of different risk levels divide the ensemble into distinct overshoot categories? We further encourage finer differentiation between categories based on their CDR assumptions, particularly for amounts of CDR required by 2050 and 2100. Such recategorization would ideally reflect key CDR differences that are made ambiguous by the current classification based on peak temperature. Finally, consider that more than two categories may be needed to distinguish scenarios along these policy-relevant dimensions.

We thank the authors for raising these questions and appreciate the opportunity to provide comments on how the modeling community can further inform the climate policy discussion on these issues.

Gabrielle B. Dreyfus, Chief Scientist, Institute for Governance & Sustainable Development

Julie S. Miller, Research Associate, Institute for Governance & Sustainable Development

Alyssa Hull, Research Associate, Institute for Governance & Sustainable Development

**References**

Goldstein A., *et al.* (2020) *Protecting irrecoverable carbon in Earth's ecosystems*, Nat. Clim. Change 10(4): 287–95..

Grubert E. & Talati S. (2024) *The distortionary effects of unconstrained for-profit carbon dioxide removal and the need for early governance intervention*, Carbon Management 15(1): 1–21.

Lohmann J. & Ditlevsen P. D. (2021) *Risk of tipping the overturning circulation due to increasing rates of ice melt*, Proceedings of the National Academy of Sciences 118(9): 1-6.

Nzotungicimpaye C.-M., MacIsaac A. J., & Zickfeld K. (2023) *Delaying methane mitigation increases the risk of breaching the 2 °C warming limit*, Commun Earth Environ 4(1): 1–8.

Ramanathan, V., Xu, Y. & Versaci, A. (2021) *Modelling human–natural systems interactions with implications for twenty-first-century warming*, Nat Sustain 5: 263-271.

Reisinger A. & Geden O. (2023) *Temporary overshoot: Origins, prospects, and a long path ahead*, One Earth 6(12): 1631–1637.

Ritchie P. D. L., Alkhayuon H., Cox P. M., & Wieczorek S. (2023) *Rate-induced tipping in natural and human systems*, Earth System Dynamics 14(3): 669–683.

Zickfeld K., MacIsaac A. J., Canadell J. G., Fuss S., Jackson R. B., Jones C. D., Lohila A., Matthews H. D., Peters G. P., Rogelj J., & Zaehle S. (2023) *Net-zero approaches must consider Earth system impacts to achieve climate goals*, Nature Climate Change 13(12): 1298–1305.

---

## Author Response (AR1)

**EGUSPHERE-2024-453**
**Combined responses to the Reviews**

**Review #1 of the paper "Bringing it all together: Science and modeling priorities to support international climate policy", by Jones C.G. et al.**

**Overview**

The paper considers international climate policy needs for science and modeling out to 2030, and lays out priorities across seven areas. These range from modeling coordination in support of the assessment reports (AR) of the Intergovernmental Panel for Climate Change (IPCC), to underpinning science foci, and also the required technological infrastructure. The paper is penned by a large team of European authors involved in IPCC, the Coupled Model Intercomparison Project (CMIP) and more generally the climate science and modeling enterprise.

The focus of the paper is important. As climate change aggravates, and the stakes for climate science and policy get higher, it is critical to have clear climate science and modeling priorities, and community coordination around those, to rapidly accelerate progress. The paper is also timely, as CMIP7 and IPCC AR7 are getting underway. The commentary adds the viewpoint of a segment of the European modeling community, to a number of manuscripts on modeling strategy that have recently been published or circulated across the international modeling community.

The priorities outlined in the paper are a reasonable evolution of what's already at play, and aim at addressing some current gaps. Overall, this Reviewer agrees with the points made in this article.

A number of specific major comments/recommendations are listed below, along with some minor points.

Overall the paper is well-written and a useful contribution to on-going community discussions around the future of climate modeling and underpinning science.

*We thank the reviewer for their insightful comments and recommendations, the majority of which we agree with. Below we respond to each point raised and outline where in the revised paper we have added or modified text to address the specific concerns. The reviewer comments are in black text and our responses in blue. The revised paper is also included in this reply so the reviewers can see how our responses to their comments fit into the overall paper.*

**Major Points**

**1.1** Line 180. The type of infrastructure outlined here still reflects a linear model from modeling to services, whereby the modeling community decides what simulations to run and shares those with users. Given the rapidly evolving climate policy questions, we should prioritize the development of an infrastructure that supports co-production of information based on climate models (i.e., experiments that are responsive to the evolving needs); that supports ML/AI exploitation of both modeling data and observations to address service needs in a flexible manner.

We agree there is a need to enhance the co-production of information based on climate and Earth system models and ease the exploitation of model data. We have updated some of our text to better emphasize this. Concerning design of model experiments themselves, this is addressed at the international level in the preparation of CMIP7 etc. Our paper proposes to better integrate the chain from IAM to climate/ES and impact models. This will facilitate improved links with users of this data by generating a more internally consistent set of simulation data across the models and modelling communities involved, achieving this in a more rapid and efficient manner. Concerning model data, more flexibility and easier exploitation is certainly required, in particular the ability to support new ML/AI applications. Better integration with the impact community, as proposed, will also help better address user needs. However, a full co-production with users of the entire experimental chain is beyond what can be addressed by the infrastructure for CMIP, CORDEX and ISIMIP, and will require significant international investment and coordination to move certain parts of the IAM-CMIP-CORDEX-impact modelling chain to a more operational setting. We briefly discuss the benefits (and challenges) in moving in this direction in lines 290 to 320.

New text to address the points related to co-development and infrastructure flexibility have been added at lines 182 to 187 and in section 8.

**1.2** Lines 240-280. Indeed, the lack of consistency and disconnects in modeling across IPCC WG1-WG2-WG3, and the relevant modeling frameworks (CMIP/CORDEX/IAMS, etc.)  are major gaps that need to be addressed. The authors do a great job of explaining current shortcomings and what could be done to address them. In the recommendations (lines 270-280), it seems important to emphasize that: 1) continuing CMIP experiments is critical to the continued improvement of models and scientific understanding; 2) a common framework of protocols, forcings, evaluation metrics, etc.  is necessary across the various modeling communities to address the disconnects (e.g., between CMIP and CORDEX); common workflows are necessary but not sufficient; 3) the recommended service oriented/quasi-operational activity and CMIP/CORDEX science activities should be well-connected, i.e. the service activity (from global to local scales) should be a purposeful spin-off of the CMIP, and service needs should be driving CMIP science.

Agreed and we have endeavoured to stress the importance of these suggestions with new text at lines 290 to 320 and lines 601 to 604.

**1.3** Lines 285-300. This is a great set of questions to illustrate the climate/Earth system modeling needs to inform climate mitigation. It is increasingly clear that changes in aerosols are a critical factor in the Earth's energy budget and that future mitigation pathways need to consider aerosol/air quality policies.  Hence, I recommend explicitly mentioning a question about understanding the interplay of GHGs and aerosols in determining future climate mitigation pathways.

Completely agree. Thanks for pointing this out. A major omission on our part. Some new text added to address this at Lines 361-362 and 734 to 746

**1.4** Lines 305-315. In addition to carbon interventions through AFOLU, there are many other types of carbon dioxide removal (CDR) methodologies that are being proposed, including enhancing the ocean carbon uptake (mCDR). Hence, ESM should also capture relevant ocean carbon cycle processes not just land CDR processes. I recommend the discussion be amended in this regard.

Agreed and we have added some text at lines 384 to 393 to address this.

**1.5** Lines 350-370. This is a great list of Earth system interactions to be examined. I would add "humans-climate/ES interactions" to this list, as humans are the current major driver of change at this point. Given the discussions regarding intentional climate interventions, as we learn about how air quality policies are affecting climate, as we are looking to diverse solutions

to the climate, biodiversity and socio-economic crises, it becomes increasingly important to factor in humans in ways that are more advanced than what we have thus far. I would recommend the authors make an additional effort in this regard across the paper.

We agree that more emphasis needs to be placed on human – Earth system interactions. Particularly with respect to unintended consequences arising from human actions designed to address climate or air quality mitigation and/or regional to national scale adaptation. We have added some text to address this at lines 436 to 440 and lines 461 to 464.

**1.6** Lines 375-390. The focus of this section on improving regional climate information is appreciated, however the discussion could be improved in several respects. 1) Global variable resolution models and two-way nested global models, both achieving resolutions comparable to regional climate models, are now a reality. These should be mentioned along side more traditional regional climate models. 2) The paper mentions models "..all running in a tightly linked framework..". Indeed, this simulation workflow is needed. What's also needed but not mentioned, is a common, model-agnostic, evaluation framework, with metrics and standards applicable across various modeling methodologies; this is increasingly important as the types of modeling methodologies diversify, e.g. with the advent of AI-based models. 3) It is striking that ML/AI is not mentioned in this discussion. It is certainly a promising tool e.g., to get to higher resolution information from lower resolution models, again duly vetted as any other modeling tool.

Thanks for underlining those aspects. We now explain in the paper that the term "regional downscaling" is intended to cover all physics-based dynamical models that aim to represent at fine-scale the climate of a specific region of the world, whatever the technical choice. This includes limited-area models (LAM), variable-resolution GCMs (VRGCM) or two-way coupled systems and possibly very-high-resolution (mainly atmosphere-land) GCMs if they target the study of the regional to local climates. Our text is not meant to be specific to the LAM approach. We would like to stress that VRGCMs have contributed to CORDEX since its inception in 2009. We have added some text to clarify this in lines 472 to 482.

Concerning point 2) we agree on the need for a common and method-agnostic evaluation framework across global models and all forms of regional downscaling. We have added some text (lines 489 to 493) to address this point.

Concerning point 3), we now make it clear that ML-based techniques we view as an important and growing component of the umbrella term statistical downscaling (lines 480 to 482) and agree that ML-based approaches need to be carefully evaluated along with, and against, more traditional statistical and dynamical downscaling approaches (lines 489 to 493)

**1.7** Line 450. This is a good description of the opportunity provided by CMIP7 to explore using higher resolution in a balanced way, considering also other important lines of research. What I see missing, is an explicit discussion of the value of model diversity, as we try to gauge uncertainty. If we had a perfect way to model climate processes, we could forgo that. But there are still significant uncertainties and errors and so model diversity continues to be crucial.

We agree that emphasizing the continuing need for model and parameterization diversity is important and have modified our text to address this; lines 565 to 571

**1.8** Lines 470-480. I generally agree with the recommendation of creating a tighter linkage between the global climate modeling (CMIP) and regional modeling (CORDEX). However, we now have global models that can get to convective-scale resolutions at the regional level, via 2-way nesting and variable resolution. These types of models would inherently bring more consistency across spatial scales and could naturally allow for a convergence of the global and regional modeling communities. Some additional discussion of these diverse opportunities is necessary here.

As stated in our response to comment 1.6, we have now made it clear that "regional downscaling" as used in our paper, refers to the full range of approaches used to generate high-resolution realizations of climate over a specific region. We agree that variable resolution GCMs and (potentially) atmosphere-land only global models running at km-scale, for timeslice experiments, may well become a reality in the coming decade. We have added some text to cover this at lines 472 to 478.

**1.9** Lines 520-535 A few considerations: 1) "Digital Twins of the Earth" (DTE) are, thus far, primarily focused on atmospheric dynamics rather than Earth system modeling; 2) The value of DTE for climate modeling will need to be evaluated with the same frameworks/metrics that we use for any new modeling tools; this value remains to be proven; 3) DTE for climate will need to explore uncertainties (modeling/data choices, internal variability, forcings, etc.); this is a "must" for any application to climate risk evaluation, as discussed in section 7; 4) it is not unique to DTE to attain km-scale resolution and be responsive to user needs. Global-convection resolving models and variable high-res climate models are a reality. The discussion of DTE should touch on all these points, to illustrate how DTE is an interesting concept but its application to climate modeling is quite aspirational at this point.
Thanks for making these completely fair points. We have added some text in lines 684 to 693 to highlight these points.

**1.10** Lines 750-795. This is a very well-written description of the status of the modeling infrastructure and the gaps that need to be addressed. One point that seems worth emphasizing: we need to evolve to an infrastructure that allows greater co-production of information between the modeling centers and the users, and greater flexibility.
We agree and have modified the text to emphasize this point; see lines 182 to 187 and in section 8.

**1.11** Section 9. A few things seem worth addressing here and throughout the paper: 1) Climate policy requires that we project out at least 100 years from the present time. It is critically important that CMIP7/AR7 cover at least the period until 2130. 2) Aerosols should be mentioned explicitly as something that we need to study in conjunction with changes in the carbon budget, as we examine future climate mitigation pathways. 3) It is notable that the paper does not touch on modeling of solar radiation modification. As these types of intervention/geoengineering are being proposed, it seems important to document potential impacts and uncertainties. 4) The paper does a good job of discussing the importance of quantifying uncertainty and how it flows across modeling systems. However, the importance of understanding predictability (what we think we can predict and why) is not explicitly addressed. 5) Lastly, the recommendation to be more inclusive of global South scientists is meritorious and could be even more convincing if the paper were to include views from co-authors from that region.
Points 1, 2 and 3 Agreed and we have modified or introduced new text in lines 1001 to 1031.

Point 4: We have mentioned this in lines 1055 to 1057. We wish to point out that we did not intend covering the topic of seasonal to decadal climate prediction. Rather this paper is mainly concerned with (uninitialized) projections. This is why we have tried to carefully use the word "projection" rather than "prediction" throughout. Although, we agree that understanding what is potentially predictable or not is of great importance.

Point 5: We fully agree, and you are right that we should have made a greater effort to engage with Global South scientists at the outset of this paper. The co-authors of this paper, and the recommendations therein, to an extent are the result of discussions we have had in a number of EU-funded research projects over the past ~5 years, and therefore primarily reflect our collective viewpoint. Within that collective opinion, there is a strong desire for our (European

and national) funding agencies and governments to recognise the need (and benefit) of funding Global South climate research. A number of co-authors have worked quite extensively with Global South scientists in international activities like; CMIP, CORDEX, ISIMIP, AgMIP, VIACS over the past 2 or more decades, and therefore have some insight into what might help. That said, you are completely right that we cannot (and should not) speak for Global South scientists. Rather we should do what we can to ensure their voices can be fully heard at relevant international fora. Our hope is that this piece might stimulate European funding agencies to ensure they provide funding and the practical means for Global South scientists to engage with European scientists on long-term projects, ideally structured around international efforts, such as those being pushed by WCRP. Examples would include CMIP, CORDEX, ISIMIP and AgMIP. It is probably a bit late in the day for us to invite Global South scientists to be co-authors on this paper. We have therefore made it clear the views expressed are opinions of this group of European researchers. We hope this paper may allow us to engage more strongly with Global South scientists in the future.

**Minor Points**

**1.12** Abstract. It seems important to add an upfront qualifier, that this is a perspective from a group of European authors.

This has been added, though in the introduction (lines 194 to 200) as we feel it fits better there than in the abstract, and in the summary (line 1082).

**1.13** Line 75. The stated time horizon for the priorities outlined in the paper is 6 years, out to 2030. Given what is outlined, this does not seem realistic given the inertia in the enterprise and also the type of recommendations that are provided. We understand the desire to be relevant to the CMIP7/AR7 cycle but it would be more realistic to talk about a 10-year time horizon.

Agreed and we have modified the time horizon to be "the coming decade".

**1.14** Figure 1. This figure could be improved so that it is readily understandable. The boxes could identify the primary function in plain language and add the acronym in smaller font, e.g. "Global Coupled Modeling – CMIP", "Regional Climate Modeling - CORDEX", etc.

Figure has been modified to address these suggestions

**1.15** Throughout the paper, acronyms should be spelled out. For instance, in lines 540, 585 what are EffCS, TCR, TCRE?

As far as we can tell, every acronym (apart from IPCC and UNFCCC, which we think do not need defining) is defined the first time it is referred to.

**1.16** Line 450. "..and for supporting climate change adaptation." Such references should be amended throughout the paper to also include mitigation, where it is applicable.

Done

**1.17** Line 590. "For RCMS too short evaluation runs.." This is another place to iterate the need for a common protocol/evaluation framework for regional scale modeling, agnostic of the modeling tool, whether it is a GCM, RCM or an AI based model.

We have added text to emphasize the need for this at lines 489 to 493 and lines 764 to 768

**Review #2 of "Bringing it all together: Science and modeling priorities to support international climate policy" by Jones et al.**

The manuscript is an opinion article presenting a large number of authors' views on the (past and) present state of the international Earth system / climate modeling efforts, including assessment, impact, regional, etc. modeling and their recommendations on a number of priority research areas moving forward. As such, it will be another addition to the recent surge of similar opinion articles (some cited already in the manuscript). While I will respect the authors' opinions in my review because of the nature of the manuscript, I will offer a few comments and suggestions for authors' consideration below.

We thank the reviewer for their insightful comments and recommendations, the majority of which we agree with. Below we respond to each point raised and outline where in the revised paper we have added or modified text to address the specific concerns. The reviewer comments are in black text and our responses in blue. The revised paper is also included in this reply so the reviewers can see how our responses to their comments fit into the overall paper.

**Reviewer comments and recommendations**

2.1 As I indicated above, there has been a recent surge of similar opinion pieces which advocate for similar approaches going forward based on lessons learnt from the previous related efforts. The current effort is certainly more comprehensive than the others, but it will be useful to mention these recent reviews / opinions up front to provide the context and the need for the present manuscript, essentially answering why the community needs another such piece.
We have referred to these earlier perspectives in our paper and outlined how we take what is suggested in these articles a bit further in terms of details, scope or ambition. In particular, see lines 290 to 320.
https://www.nature.com/articles/s41558-023-01909-9
https://www.nature.com/articles/s41558-023-01849-4
https://agupubs.onlinelibrary.wiley.com/doi/full/10.1029/2023AV001086
https://doi.org/10.1093/acrefore/9780190228620.013.933

2.2 As far as I can tell, all the co-authors of this manuscript are from European institutions. So, this article represents ONLY a "European" view of how these truly international – not just European – efforts need to be done. International community is not just Europe! This should be made clear.
We have made this clear in the introduction (lines 194 to 200) and in the summary section (line 1037)

2.3 The title may be interpreted to imply that the community does Earth system and climate modeling in service to climate policy only. I do not think that this is the intention of the authors. It is important to clarify that Earth system science stands on its own and a subset of related efforts serve the international climate policy.
Fair point. We propose a modified title:
Bringing it all together: Science priorities for an improved understanding of Earth system change and to support international climate policy.

2.4 We are in the era of co-development, co-design, co-planning, co-analysis, co-etc. of all these efforts with the communities that are impacted by climate change. While the article mentions mitigation, adaptation, etc. and related efforts, given the author list, all of these views and recommendations do not really reflect the views of impacted communities and the Global South for that matter. To avoid "we know the best for your community" perception, please be mindful of this and add caveats, acknowledging that

these are only "European" views and recommendations and that they may not reflect the true needs of the impacted communities.

Agreed and we have added text stressing the importance of co-production, analysis etc at lines: 183, 296, 922 and 928.

Agreed and caveats have been added to stress these are views from a group of European researchers only: Lines 194 to 200 and Line 1082.

2.5 Many of the challenges and issues covered in the article are not new. They have not been addressed for many reasons – some are discussed in the manuscript. The article states a target date of 2030 to accomplish some of these while starting some progress on the others. This is a rather tall order. It will be good to discuss what changed over the last few years that make the authors think that there can be significant progress on these challenges, especially noting that CMIP7 timeline is rather short with quite a few of the recommendations need to start very soon, if not now.

We have changed our proposed timeline to the "coming decade". We have also emphasized that some work may deliver into IPCC AR7, but that these timelines are very tight. The CMIP7 science MIPs are likely to continue through to 2030, so we have highlighted that much of our proposed work will deliver into "an improved understanding of the Earth system and Earth system change", with subsequent benefits for climate policy arising later e.g. IPCC AR8 and the 2033 Global Stocktake.

We have added some new text (lines 325 to 352) that outline aspects we feel have changed the situation over the past ~5-10 years that make us feel a number of key areas we discuss are now ripe for rapid development over the coming decade.

2.6 Related to #5 above, I suggest including a review of what the real and perceived impediments have been to date to accomplish the discussed recommendations and a discussion of what the impediments are going forward. Otherwise, I fear that this piece will be another "opinion" piece to be added to the existing ones without really addressing such impediments in a concrete way.

We have tried to address this request (not the easiest one to address) through new text at lines 325 to 352 and lines 995 to 999.

2.7 The manuscript has the feel of written by several authors. I suggest that the lead authors go over the acronyms, definitions, etc. carefully, making the manuscript more coherent. Machine learning is mentioned / discussed as a way forward in many of the sections with similar sentences. Should it have its own section and discussion, perhaps at the end, tying things together? There are also quite a few sentences that are long with multiple groups of a few words separated by commas. Such sentences are rather difficult to parse and understand. An example is the sentence on lines 526-529. Please rephrase these sentences.

We have gone through the manuscript and tried to homogenize the style and make the paper read more easily. All acronyms (except IPCC and UNFCCC which we feel do not need defining) are defined the first time they are mentioned. We have tried to break up excessively long sentences (as much as possible!).

On the topic of Machine Learning and AI. We agree this is potentially transformative for the science of Earth system change. Where we feel it can make solid contributions over the coming decade to the challenges we identify, we have tried to indicate this. That said, with respect to (i) generation of emission scenarios, (ii) Earth system modelling and global ES projections and (iii) climate change impact modelling, we do see the potential for ML and AI to make significant contributions, but nevertheless feel that the more established (physics/Navier-Stokes based) approaches to modelling still

have a great deal to offer over the coming decade. We have tried to be careful not to write this paper as a review of ML/AI potential or as a perspective piece on ML and AI. Such papers are appearing in the literature, and we refer to a number of these. We prefer not to have a specific section on ML and AI but rather leave that to other papers more specifically dedicated to that topic.

---

## Author Response (AR2)

Dear Editor

Many thanks for the constructive and positive comments on our perspective paper (copied below).

To respond to these, we have added a new paragraph into the paper at **lines 193- 206**, further detailing that a number of highly relevant perspective papers have appeared over recent years, touching on areas common to our perspective piece. We encourage the readers to read our piece in the light of these other perspectives and emphasize our piece aims to discuss priorities across the entire chain of activities that deliver scientific knowledge into the international and national policy arenas.

In addition, as we very much agree with the need for a balanced approach to future model development, as outlined in https://egusphere.copernicus.org/preprints/2024/egusphere-2024-20/ and recommended by the reviewer, we have strengthened our statements around the need for increased collaboration between physics/process-based modelling and hybrid/Machine Learning approaches at **lines 93 to 96 and lines 683 to 686**.

We hope that these changes are sufficient to allow the paper to be accepted for publication.

Best wishes

Colin Jones (Prof)

....

**Reviewer comments:**

One remaining recommendation is to include a short mention/discussion of other relevant opinion pieces. The fact that this opinion is one of several that have recently appeared should be recognized in the introduction; a short discussion of major similarities could be included towards the end of the paper. Examples of relevant opinions include:
https://egusphere.copernicus.org/preprints/2024/egusphere-2024-20/
https://doi.org/10.1029/2023EF004187
Discussing common threads, especially if they are coming from different parts of the community, would add weight to the recommendations included in this paper.

**Editors comment:**

Additional private note (visible to authors and reviewers only):
Please address Reviewer 1's suggestions, in particular, reference to recent relevant work.